# FAM3D is essential for colon homeostasis and host defense against inflammation associated carcinogenesis

Weiwei Liang[1,2,11], Xinjian Peng[1,11], Qingqing Li[1], Pingzhang Wang[1], Ping Lv[1], Quansheng Song[1], Shaoping She[1], Shiyang Huang[1], Keqiang Chen[2], Wanghua Gong[3], Wuxing Yuan[4], Vishal Thovarai[5], Teizo Yoshimura [6], Colm O'huigin[2], Giorgio Trinchieri [2], Jiaqiang Huang[2,7], Shuye Lin[7], Xiaohong Yao[8], Xiuwu Bian [8], Wei Kong [9], Jianzhong Xi[10], Ji Ming Wang[2✉] & Ying Wang [1✉]

The physiological homeostasis of gut mucosal barrier is maintained by both genetic and environmental factors and its impairment leads to pathogenesis such as inflammatory bowel disease. A cytokine like molecule, FAM3D (mouse Fam3D), is highly expressed in mouse gastrointestinal tract. Here, we demonstrate that deficiency in *Fam3D* is associated with impaired integrity of colonic mucosa, increased epithelial hyper-proliferation, reduced anti-microbial peptide production and increased sensitivity to chemically induced colitis associated with high incidence of cancer. Pretreatment of *Fam3D*−/− mice with antibiotics significantly reduces the severity of chemically induced colitis and wild type (WT) mice co-housed with *Fam3D*−/− mice phenocopy *Fam3D*-deficiency showing increased sensitivity to colitis and skewed composition of fecal microbiota. An initial equilibrium of microbiota in cohoused WT and *Fam3D*−/− mice is followed by an increasing divergence of the bacterial composition after separation. These results demonstrate the essential role of Fam3D in colon homeostasis, protection against inflammation associated cancer and normal microbiota composition.

[1] Department of Immunology, School of Basic Medical Sciences and NHC Key Laboratory of Medical Immunology, Peking University, Beijing 100191, P. R. China. [2] Cancer and Inflammation Program, Center for Cancer Research, National Cancer Institute at Frederick, Frederick, MD 21702, USA. [3] Basic Research Program, Leidos Biomedical Research, Inc, Frederick, MD 21702, USA. [4] Microbiome Sequencing Core, Leidos Biomedical Research, Inc, Frederick, MD 21702, USA. [5] Basic Science Program, Leidos Biomedical Research, Inc., Frederick National Laboratory for Cancer Research, Frederick, MD 21702, USA. [6] Department of Pathology and Experimental Medicine, Graduate School of Medicine, Dentistry and Pharmaceutical Sciences, Okayama University, Okayama 700-8558, Japan. [7] Cancer Research Center, Beijing Chest Hospital affiliated to Capital Medical University, Beijing Tuberculosis & Thoracic Tumor Research Institute, Beijing 101149, P. R. China. [8] Institute of Pathology, South-west Hospital and Cancer Center, Chongqing, P. R. China. [9] Department of Physiology and Pathophysiology, School of Basic Medical Sciences, Peking University, Beijing 100191, P. R. China. [10] Department of Biomedicine, College of Engineering, Peking University, Beijing 100871, P. R. China. [11] These authors contributed equally: Weiwei Liang, Xinjian Peng. ✉email: wangji@mail.nih.gov; yw@bjmu.edu.cn

nflammatory bowel disease (IBD) is a complex and multifactorial disorder that includes Crohn's disease and ulcerative colitis (UC), both characterized by chronic relapsing inflammation and intestinal epithelium damage associated with increasing risk of colorectal cancer[1–4]. UC is a major form of IBD and its etiology remains unclear. Mounting evidence suggests impaired barrier function of colon mucosa as a critical factor contributing to UC pathogenesis[5].

Mammalian gut harbors a complex microbial ecosystem resulting in the establishment of a mutually beneficial relationship between the microbiota and host, which is critical for maintaining gut homeostasis[6]. The intestinal epithelial cells (IECs) constitute the first physical and immunological protective barrier against pathogens[5]. For example, the secretory IECs such as goblet cells produce mediators present in the intestinal lumen, including polymeric gel-forming mucin MUC2 that forms a protective mucus layer and distinct families of antimicrobial peptide-like C-type lectins (such as the regenerating islet-derived protein (REG) family), defensins and resistin-like molecule-β. These molecules participate in the normal separation of microbiota from colon tissue and prevent mucosal inflammatory responses induced by invasive abnormal microbiota and enteric pathogens[7–9]. Damage to the intestinal epithelial barrier results in dysbiosis in the gut, which activates innate immune responses and induces inflammatory disorders[10,11].

FAM3D (Fam3D in mice, family with sequence similarity 3, member D) was originally identified as a member of a cytokine-like family with a four-helix-bundle structure similarity, consisting of four gene products, FAM3A, FAM3B, FAM3C, and FAM3D[12]. FAM3D is a gut-secreted protein reported to be regulated by nutritional status with unknown function[13]. FAM3D (Fam3D) was subsequently identified as a host-derived endogenous chemotactic agonist for the G-protein coupled receptors, namely formyl peptide receptor (FPR)1 and FPR2 (mouse Fpr1 and Fpr2)[14], suggesting its proinflammatory properties[15]. However, the role of FAM3D (Fam3D), constitutively and abundantly expressed in the gastrointestinal tract, in the homeostasis and pathological conditions in the colon remains unclear. In this study, by using mice deficient in the gene for Fam3D, we demonstrate Fam3D as a key contributor to colon homeostasis, protection from inflammation-associated carcinogenesis and microbiota balance.

## Results

**High-level expression of FAM3D (mouse Fam3D) in mouse gastrointestinal tract**. FAM3D is expressed in human and mouse gastrointestinal tissues as a gut-secreted protein[13]. Analysis of normal mouse tissue cDNA panel revealed abundant expression of Fam3D in the gastrointestinal tract, in particular in the colon, with moderate expression in the mesenteric lymph nodes (MLN), lung, thymus, and spleen, but not in other tissues (Fig. 1a). Western blot confirmed high-level Fam3D protein in the mouse gastrointestinal tract (Fig. 1b), with an amount around 90 ng/30 µg total colon tissue (Supplementary Fig. 1a).

Immunohistochemistry revealed that Fam3D was constitutively expressed in epithelial cells with a prominent villus and crypt expression pattern (Fig. 1c, Supplementary Fig. 1b) and immunofluorescent staining showed the presence of Fam3D in the apical regions in crypt epithelial cells (Supplementary Fig. 1c). We isolated IECs from normal mouse colon and found abundant expression of Fam3D mRNA as measured by real-time PCR (Fig. 1d), which was consistent with high levels of Fam3D protein in colonic IECs as confirmed by Western blot (Fig. 1e). Also, dual immunofluorescent staining showed the production of Fam3D by colonic epithelial cells (EpCAM-positive cells) (Supplementary Fig. 1d). In intestinal organoids, we detected Fam3D in crypt structures that recapitulated native intestinal epithelium, with

higher expression level in the organoids stimulated by DAPT, a Notch inhibitor, to induce the differentiation of cultured Lgr5+ intestinal stem cells as measured by Western blot and confocal microscopy (Supplementary Fig. 1e, f). We therefore confirm that Fam3D is constitutively expressed by colonic epithelial cells.

**Increased spontaneous colitis and global inflammation in Fam3D−/− mice**. To examine whether endogenously produced high level of Fam3D actively participates in the pathophysiological processes, in particular the homeostasis, in the colon, we generated Fam3D−/− mice. The colons of naive Fam3D−/− mice exhibited crypt hyperplasia with the length of crypts significantly elongated compared with the WT counterparts (Fig. 1f). This was associated with increased Ki67-positive proliferating epithelial cells in the colonic mucosa of Fam3D−/− mice (Fig. 1g). In WT mice, Ki67-positive proliferating cells were abundant at the bottom of the colon crypts, but were reduced or absent in epithelial cells moving toward the top of the crypts. By contrast, Ki67-positive cells observed in Fam3D−/− mice showed either a disorganized pattern, mainly located in the middle and bottom of the crypts or having a diffused distribution.

We then analyzed the stem cell compartment of Fam3D−/− mouse colon. There was no difference in the numbers of Lgr5+ cells in the colon of WT and Fam3D−/− mice as measured by immunofluorescent staining (Supplementary Fig. 2a), or stem cell-related genes, such as Axin2, Olfm4, Smoc2, Msi1, and Ascl2. as measured by RNA-seq (Supplementary Fig. 2b, c). Therefore, deficiency in Fam3D did not affect the generation of colon epithelial stem cells. We further analyzed the stem cell compartment by using the organoid culture. The formation and size of organoids from isolated crypts did not show the difference between WT and Fam3D−/− mice (Supplementary Fig. 2d, e). These results confirm that Fam3D is unlikely involved in the renewal of the colonic mucosal layer.

Figure 1h shows the infiltration of CD3+ T and B220+ B cells in the colonic epithelium of Fam3D−/− mice, suggesting the presence of a low-level inflammatory process. The number of F4/80+ macrophages in both WT and Fam3D−/− mouse colon was similar. No acute inflammation with neutrophil infiltration was detected in Fam3D−/− mouse colon. The low level of lymphocytic inflammation was confirmed by proinflammatory cytokine gene expression measured by real-time PCR (Fig. 1i).

We then examined the pathology of 1-year-old mice and found an exacerbated crypt hyperplasia in Fam3D−/− mice with neutrophil infiltration and invasive glands (Fig. 1j), as well as the loss of differentiated epithelial cells (Fig. 1k), with increased leukocytes in the colonic epithelium. Significantly increased CD11b+ myeloid and CD3+ T cells were isolated from the colon of Fam3D−/− mice compared to WT control mice (Supplementary Fig. 3a, b). In addition, there were increased weight of colons from Fam3D−/− mice (Supplementary Fig. 3c). Moreover, an enlargement of the spleens and increased circulating inflammatory leukocytes were observed in Fam3D−/− mice, indicating the presence of global inflammatory responses (Supplementary Fig. 4a–c). These results indicate spontaneous colitis due to long term Fam3D deficiency.

**Reduced homeostatic molecules in the colon epithelium of Fam3D−/− mice**. We then characterized colonic epithelial changes by whole-transcriptome RNA-seq, which revealed a reduction of 52 genes with an increase of 24 genes in Fam3D−/− mice (Supplementary Fig. 5a, b). Some of the reduced genes encoded antimicrobial peptides (Reg3b, Reg3g, and Saa3) (Fig. 2a) and others were associated with responses to type I and type II IFNs (Iigp1, Ifi44, Ifi47, Ifit2, etc.) (Supplementary Fig. 5c). Real-

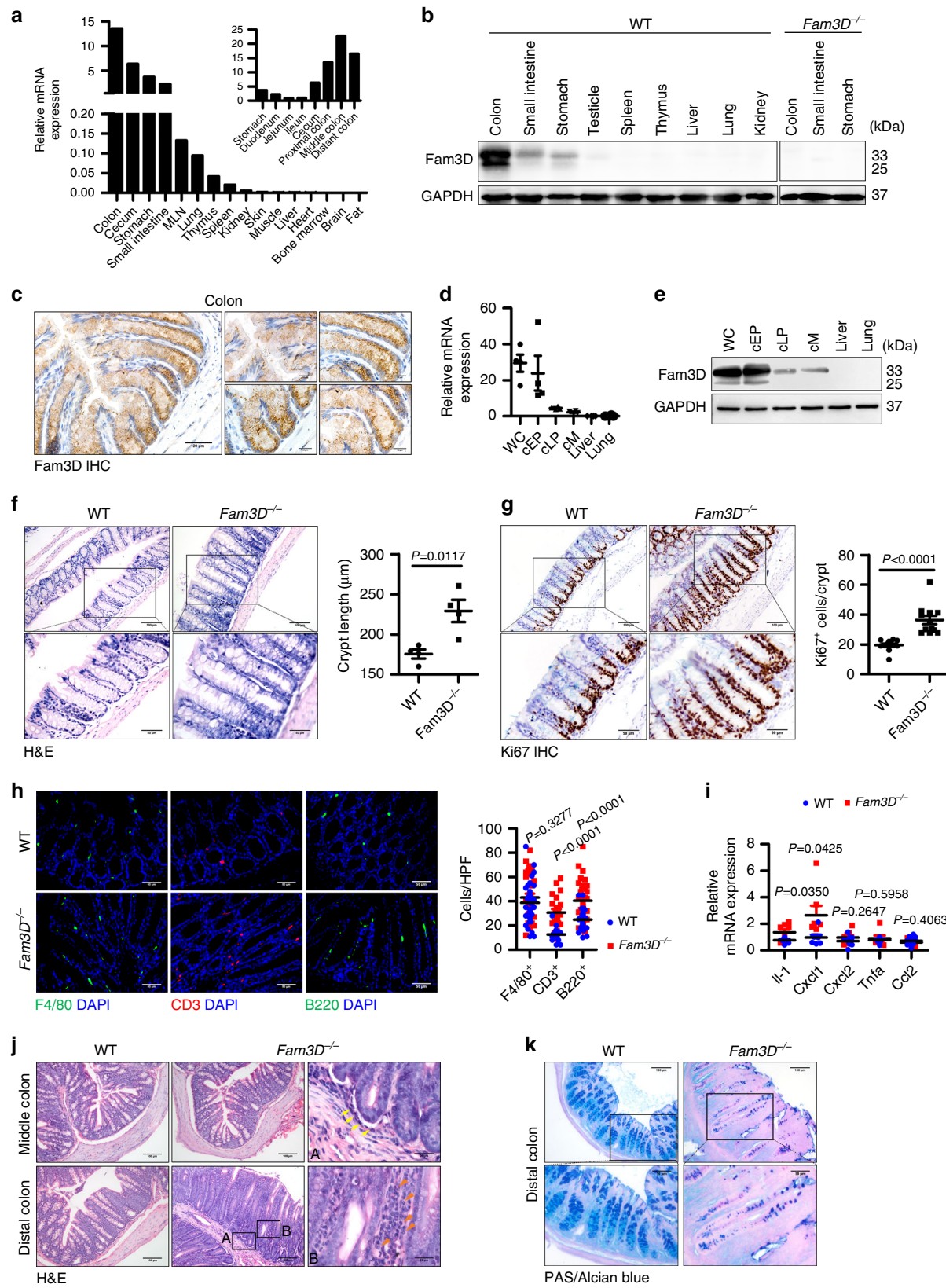

time PCR validated reduction of mRNA levels for *Reg3b*, *Reg3g*, *Saa3*, *Defb1*, and *Defa2* (Fig. 2b) with decreased Reg3γ protein in the colon of *Fam3D*$^{-/-}$ mice (Fig. 2c). Since activation of Reg3 members and production of interferon γ in the intestinal tract are important innate responses to bacterial colonization, decreased

expression of these molecules suggests the possibility of impaired homeostatic state of the colon in *Fam3D*$^{-/-}$ mice.

**Defective mucosal integrity in *Fam3D*$^{-/-}$ mouse colon.** We next investigated the type of colonic epithelial cells affected by

**Fig. 1 Constitutive expression of Fam3D in mouse gastrointestinal tract and loss of Fam3D leads to spontaneous colitis. a** Fam3D mRNA expression in normal mouse tissues measured by real-time PCR. MLN: Mesenteric lymph nodes. **b** Mouse Fam3D protein expression in normal tissues measured by Western blot. Tissues from $Fam3D^{-/-}$ mice were used as a negative control. **c** Representative immunohistochemical staining for Fam3D in mouse colon. Brown color indicates Fam3D. Scale bar = 20 μm; Insert scale bar = 10 μm. **d** Fam3D mRNA expression in colon fragments measured by real-time PCR. WC: whole colon. cEP: colonic epithelial cells. cLP: colonic lamina propria cells. cM: colonic muscles. **e** Mouse Fam3D protein expression in colon fragments measured by Western blot. WC: whole colon. cEP: colonic epithelial cells. cLP: colonic lamina propria cells. cM: colonic muscles. **f** Distal sections of colons obtained from 4 WT or $Fam3D^{-/-}$ mice. H&E sections showing the length of colonic crypts (11 to 20 crypts from each colon). $n = 4$ in each group. Scale bar = 100 μm; Insert scale bar = 50 μm. **g** Representative immunochemical staining of Ki67 in the distal sections of the colon from WT or $Fam3D^{-/-}$ mice. $n = 4$. Data is presented as the mean ± SEM. Scale bar = 100 μm; Insert scale bar = 50 μm. **h** F4/80-positive macrophages (green, left panel), CD3-positive lymphocytes (red, middle panel) or B220-positive lymphocytes (green, right panel) infiltrating the colon of WT and $Fam3D^{-/-}$ mice in homeostatic state. Scale bar = 50 μm. Positive cells in 10 high-powered microscopic fields (HPF) were counted in six sections from four independent mice from each group. Data is presented as the mean ± SEM per HPF. **i** The expression of proinflammatory cytokine genes including Il-1, Cxcl1, Cxcl2, Tnfa, and Ccl2 examined by real-time PCR, $n = 7$. **j**, **k** H&E and PAS/Alcian blue staining of samples from 1-year-old mice. Data is representative of one experiment repeated three independent times. Data is presented as the mean ± SEM. Statistical significance was determined by unpaired, two-tailed Student's t test.

---

*Fam3D* deficiency. We examined the profile of signature transcripts[16] using a published single-cell RNA sequencing results of intestinal epithelial cells via GSEA analysis, which revealed that the goblet cell and enteroendocrine cell signature were specifically enriched in $Fam3D^{-/-}$ mice (Fig. 2d, Supplementary Fig. 5d), suggesting enhanced goblet cell activation and/or increased representation of this lineage in the colonic epithelial, as well as enteroendocrine cells. Double immunofluorescent staining showed that Fam3D was contained in MUC2-positive goblet cells (Fig. 2e), and in CHGA-positive enteroendocrine cells (Supplementary Fig. 5e). Since the loss of Fam3D led to increased goblet cell signature, we characterized goblet cell function in $Fam3D^{-/-}$ mice. PAS and Alcian blue staining showed a significant level of goblet cell hypotrophy and hypoplasia, with an increased number and size of goblet cells (Fig. 2f). PAS-positive goblet cells, normally found in the apical and central parts of the crypts, were present at the base of the crypts in the colon of $Fam3D^{-/-}$ mice, with the absence of Alcian blue single-positive mucins. Alcian blue positive acidic mucins, as indicators of colonic epithelial cell secretory function[17], mainly composed of sulfomucin and sialomucin[18], were reduced in the colon of $Fam3D^{-/-}$ mice. HID/Alcian blue staining showed increased ratio of sulfomucin and sialomucin in the colon of $Fam3D^{-/-}$ mice (Fig. 2g). These goblet cell changes indicate altered mucin production with possible retention in the colon of $Fam3D^{-/-}$ mice.

We subsequently examined the transcriptional regulation of goblet cell-specific proteins including *Muc2*, *Muc3*, *Muc5ac*, and *Muc13* with defined roles in intestinal homeostasis. MUC2 is a gel-forming mucin and the main component of the intestinal mucus layer, while MUC3, MUC5AC, and MUC13 are surface-bound mucins involved in signaling and tumorigenesis. No reduction in the protein level of MUC2 (Supplementary Fig. 6a), nor in any goblet cell-specific protein transcript levels was detected in $Fam3D^{-/-}$ mice (Supplementary Fig. 6b), suggesting that the deficiency in mucus production in $Fam3D^{-/-}$ mice is not due to reduced transcription. However, there was a progressive reduction in the thickness of mucus layer of distal colon in $Fam3D^{-/-}$ mice (Fig. 2h), with a greater reduction in the inner mucus layer but without apparent bacterial invasion (Fig. 2i).

To further examine the development of colon crypts, we found that elongated crypts in $Fam3D^{-/-}$ mice were not present at birth but appeared progressively with aging. Goblet cell dysfunction was manifested first as a shift from the production of acidic mucin toward neutral mucin in the colonic epithelium of newborn $Fam3D^{-/-}$ mice, followed by hyperplasia and a continuous mucin shift with aging (Fig. 2j, k). Therefore, the goblet cell dysfunction in the colon precedes the elongation of crypts that results in mucus retention over time.

**Increased severity of DSS-induced acute colitis in $Fam3D^{-/-}$ mice.** Based on the prominent expression of Fam3D in mouse colon, we assessed whether the loss of Fam3D would cause abnormalities. In DSS-induced colitis, $Fam3D^{-/-}$ mice showed markedly increased severity of inflammation including more rapid body weight loss, increased rectal bleeding, and diarrhea. All $Fam3D^{-/-}$ mice died by day 9 post DSS intake, in contrast to the day 12 shown by WT mice (Fig. 3a–c). To examine the healing ability of colon mucosa, mice were given 2.5% DSS for 7 days followed by normal drinking water. Although this significantly improved disease scores, $Fam3D^{-/-}$ mice still failed to recover and all died by day 9 (Fig. 3d). Histologically, $Fam3D^{-/-}$ mice treated with DSS showed shortened colon length and marked intestinal epithelial damage by day 5 with further deterioration by day 7 (Fig. 3e–g). The colon of $Fam3D^{-/-}$ mice showed reduced epithelial cell proliferation with increased apoptosis after 7-day DSS intake (Supplementary Fig. 7a). In addition, colonic mucosa in $Fam3D^{-/-}$ mice was infiltrated by a large number of leukocytes, including monocytes/macrophages, neutrophils and T cells (Fig. 3h), as confirmed by FACS analysis (Supplementary Fig. 7b, c). The protein level of proinflammatory cytokines and chemokines, such as TNF-α and CCL2, was also significantly increased (Fig. 3i). Therefore, $Fam3D^{-/-}$ mice are more highly sensitive to DSS-induced colitis indicating a critical role of Fam3D in protecting the colon from inflammatory insults.

**Increased DSS-induced chronic colitis and AOM/DSS-induced carcinogenesis in $Fam3D^{-/-}$ mice.** In a chronic model of colitis induced by lower doses of DSS, the colon length, body weight, and the survival of $Fam3D^{-/-}$ mice were also significantly reduced in comparison with WT littermates (Fig. 4a–c). $Fam3D^{-/-}$ mice contained an increased number of larger tumors in the colon (Fig. 4d). This was accompanied by the development of marked leukocytic infiltration, extensive hyperplastic and neoplastic foci, especially near the mucosal surface (Fig. 4e) in the colon of $Fam3D^{-/-}$ mice. Ki67 staining confirmed the hyperproliferative status of these lesions in $Fam3D^{-/-}$ colon compared with adjacent non-neoplastic cells (Supplementary Fig. 8a), indicating impaired repair of the colonic epithelium after injury in $Fam3D^{-/-}$ mice with increased tumorigenesis.

Further, in AOM/DSS-induced tumorigenesis, $Fam3D^{-/-}$ mice exhibited more significant weight loss, shortened colon length, and increased number of larger tumors in the colon (Fig. 4f–i). Colorectal cancer in $Fam3D^{-/-}$ mice progressed more rapidly compared with WT mice because of more numerous invasive glands in the distal colons of $Fam3D^{-/-}$ mice that invaded deeper into the mucosal wall with extensive desmoplasia with aggressive characteristics such as atypical glandular architectures[19] (Fig. 4j, Supplementary Fig. 8b). There was

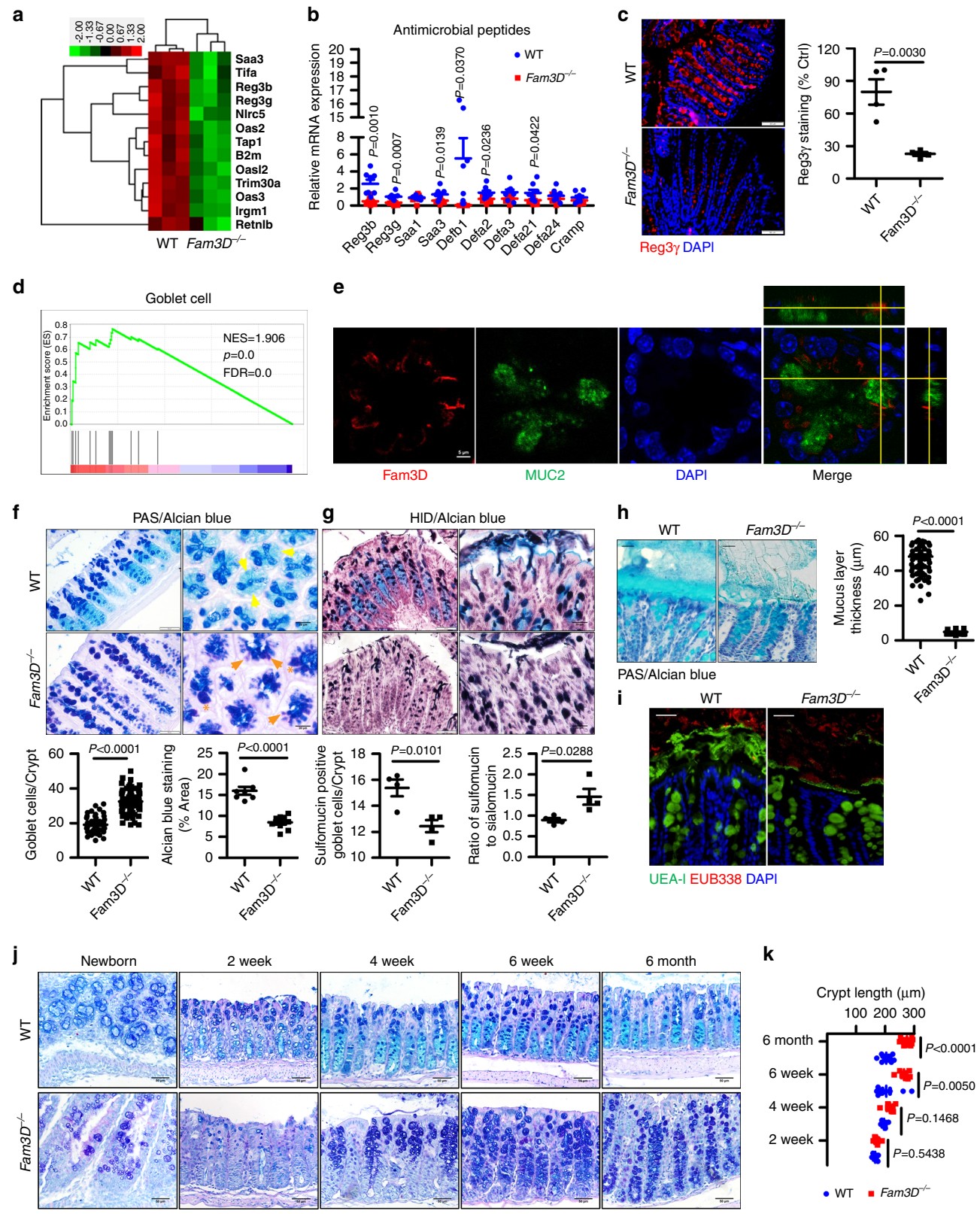

an increased level of β-catenin in proliferative crypts and invasive glands in the colon of *Fam3D*[−/−] mice (Fig. 4k). Submucosal glands displayed few positive nuclei and β-catenin was located in the membrane of epithelial cells without any nuclear accumulation (Supplementary Fig. 8c). Thus, *Fam3D* deficiency promotes inflammation-associated carcinogenesis in mouse colon, involving increased Wnt signaling as evidenced by

elevated β-catenin[20], culminating in the progression of colorectal cancer.

**Rescue of established colitis in mice by Adv-FAM3D infusion.**
To explore the potential of Fam3D as a therapeutic agent, we first tested the ability of Adv-FAM3D to transduce FAM3D in

**Fig. 2 Reduced homeostatic molecules and defective mucosal integrity in the colon of *Fam3D*⁻/⁻ mice. a** Heatmaps of differentially expressed genes enriched in host defense responses. **b** mRNA expression levels of antibacterial peptides in the colons of WT or *Fam3D*⁻/⁻ mice measured by real-time PCR. $n = 8$. **c** Reg3γ protein levels measured by immunofluorescent staining. $n = 4$ for each group. **d** GSEA analysis on *Fam3D*⁻/⁻ vs WT EBSeq log₂ FC expression data of colonic epithelial samples. **e** Immunofluorescent staining for Fam3D and MUC2 in normal colon tissue. Data is representative of one experiment repeated three independent times. **f** Double staining of Acian blue and PAS. Yellow arrow: Acian blue positive materials; orange arrow: PAS single positive cells; orange star: dilated goblet cells, goblet cells enumerated in each crypt (10–20 crypts from each colon) and pixels of Alcian blue staining areas. Alcian blue staining areas and fluorescence intensity quantified by Image J based on six distal colon sections of four independent mice from each group. **g** Staining of high iron diamine/Alcian blue, dark brown, and black were sulfomucins, whereas blue staining indicates sialomucins, sulfomucin positive cells enumerated in each crypt (10–20 crypts from each colon) and the ratio of sialomucins to sulfomucins is calculated. Data is presented as the mean ± SEM. Scale bar = 50 μm. **h** Representative Alcian blue staining of Carnoy's-fixed colonic sections. The thickness of the inner mucus layer (light blue band between luminal content and the mucosa) was quantified (5–10 fields from each colon) by Image J. $n = 8$. **i** Representative dual staining for UEA-I (immunofluorescence, green) and bacteria (fluorescence in situ hybridization, red), the latter using the universal EUB338 probe, on Carnoy's-fixed colonic sections. $n = 8$. **j** PAS/Alcian blue staining of the colons from WT and *Fam3D*⁻/⁻ mice with different ages. $n = 4$. Data is representative of one experiment repeated three independent times. **k** Quantitation of crypt length of WT and *Fam3D*⁻/⁻ mice at different ages. Data is presented as the mean ± SEM. Statistical significance was determined by unpaired, two-tailed Student's *t* test.

epithelial cells. A human colonic epithelial cell line FPCK-1-1 infected with Adv-FAM3D expressed high levels of FAM3D (Supplementary Fig. 9a). Western blotting detected FAM3D in both supernatants and cell lysates (Supplementary Fig. 9b). We then infused Adv-FAM3D into the rectum of mice and detected increased expression of FAM3D in colon tissues (Supplementary Fig. 9c, d).

To determine the protective efficacy of Adv-FAM3D on colitis, mice received triple intrarectal injections of 10⁹ PFU/mouse of Adv-FAM3D before DSS treatment (Fig. 5a). Both WT and *Fam3D*⁻/⁻ mice that received Adv-FAM3D demonstrated a significant improvement in the severity of colitis including body weight and colon length over the 7-day DSS treatment, in contrast to Adv-null treatment of mouse colon (Fig. 5b, c). Histological analysis of colonic tissue from mice with 7-day DSS treatment infected with Adv-null demonstrated a marked colitis with a totally damaged mucosa and leukocyte infiltration, but treatment with Adv-FAM3D showed a significant restoration from colitis in both WT and *Fam3D*⁻/⁻ mice (Fig. 5d). These results indicate that rectal Adv-FAM3D infusion ameliorates established colitis thus confirming the protective role of FAM3D.

**Microbiota-driven spontaneous and DSS-induced colitis in *Fam3D*⁻/⁻ mice.** Microbiota contributes to the pathogenesis of colitis[11,21,22]. To determine the role of microbiota in the colitis phenotype of *Fam3D*⁻/⁻ mice, mice were administered antibiotics, which eliminated the difference in colon crypt length between WT and *Fam3D*⁻/⁻ mice (Fig. 6a) and significantly reduced the severity of colitis in *Fam3D*⁻/⁻ mice (Fig. 6b, c). These results suggest that dysbiosis contributes to the dysfunction and increased susceptibility of *Fam3D*⁻/⁻ mice to colitis. However, compared with WT mice, *Fam3D*⁻/⁻ mice after antibiotic treatment continued to show a persistent mild degree of colitis with mucosal damage and lower degree of leukocyte infiltration on day 5 after DSS treatment (Fig. 6d). Antibiotic treatment also did not fully restore the expression of acidic mucins by epithelium in *Fam3D*⁻/⁻ mice (Fig. 6e), suggesting gut microbiota may not fully account for the colitis phenotype in *Fam3D*⁻/⁻ mice. Therefore, increased severity of colitis in *Fam3D*⁻/⁻ mice appears to be partially microbe-driven, with preexisting dysfunction of goblet cells as an important contributing factor.

**Altered composition of microbiota in the colon of *Fam3D*⁻/⁻ mice.** The ability of antibiotics to partially reduce the severity of colitis in both WT and *Fam3D*⁻/⁻ mice prompted us to investigate the composition of gut microbiota. Analysis of the α diversity of the microbiome, which defines the local bacterial

species diversity, in fecal bacterial DNA showed no significant differences in the numbers of the shared operational taxonomic units between WT and *Fam3D*⁻/⁻ mice. However, the Shannon diversity index was significantly reduced in *Fam3D*⁻/⁻ mice, suggesting decreased richness of microbiome (Supplementary Fig. 10a). In addition, Jaccard emperor-principal coordinates analyses showed that the microbiome in *Fam3D*⁻/⁻ mouse colon clustered separately from that in WT colon (Fig. 6f). Analysis of relative abundance of predominant bacterial clades showed that at the phylum level, feces from *Fam3D*⁻/⁻ mice contained expanded abundance in *Bacteroidetes* (Fig. 6g, h). At the family level, fecal microbiota of *Fam3D*⁻/⁻ mice had a higher relative abundance of *Muribaculaceae* and a lower abundance of *Ruminococcaceae* and *Deferribacteraceae* (Fig. 6i, j). Real-time PCR confirmed the dysbiosis in *Fam3D*⁻/⁻ mouse colon, including increased abundance in *Lactobacillus/Lactococcus* and *Peptostreptococcus* in Firmicutes, but decreased Segmented filamentous bacteria, *Eubacterium rectale*, *Enterococcus faecalis* in Firmicutes, and Enterobacteria in Proteobacteria (Supplementary Fig. 10b, c). These findings confirm dysbiosis in *Fam3D*⁻/⁻ mice.

**Transfer of *Fam3D*⁻/⁻ mouse phenotype to WT mice.** To obtain additional evidence for the association of dysbiosis with *Fam3D* deficiency, we cohoused WT and *Fam3D*⁻/⁻ mice followed by separation. Figure 7a showed that there was an initial equilibrium of microbiota after 4 weeks cohousing of WT and *Fam3D*⁻/⁻ mice. However, after separation, there was a progressive divergence of bacterial composition in the feces of *Fam3D*⁻/⁻ mice. After 3-month of separation, WT and *Fam3D*⁻/⁻ mice resumed colitis phenotype similar to the characteristics before co-housing (Supplementary Fig. 10d, e), suggesting that alteration of microbiota in the colon is a result of *Fam3D* deficiency.

In addition, after cohousing, the crypt length of the colon in naïve WT mice became elongated as seen in naïve *Fam3D*⁻/⁻ mice (Fig. 7b). In acute colitis, WT mice cohoused with *Fam3D*⁻/⁻ mice showed more significant body weight loss, shortening of colon length, and epithelium damage as compared with non-cohoused WT mice. However, there was no change in the severity of colitis between non-cohoused and cohoused *Fam3D*⁻/⁻ mice (Fig. 7c–e), indicating a transfer of *Fam3D*⁻/⁻ phenotype to WT mice after cohousing. Figure 7f showed a significantly different microbiota composition between non-cohoused WT and *Fam3D*⁻/⁻ mice. After 4-week cohousing, the microbiota composition in the feces from WT mice shifted toward that of *Fam3D*⁻/⁻ mice. Treatment with DSS for 3 days revealed a closer microbiome composition between cohoused WT and *Fam3D*⁻/⁻ mice. The differences in the composition

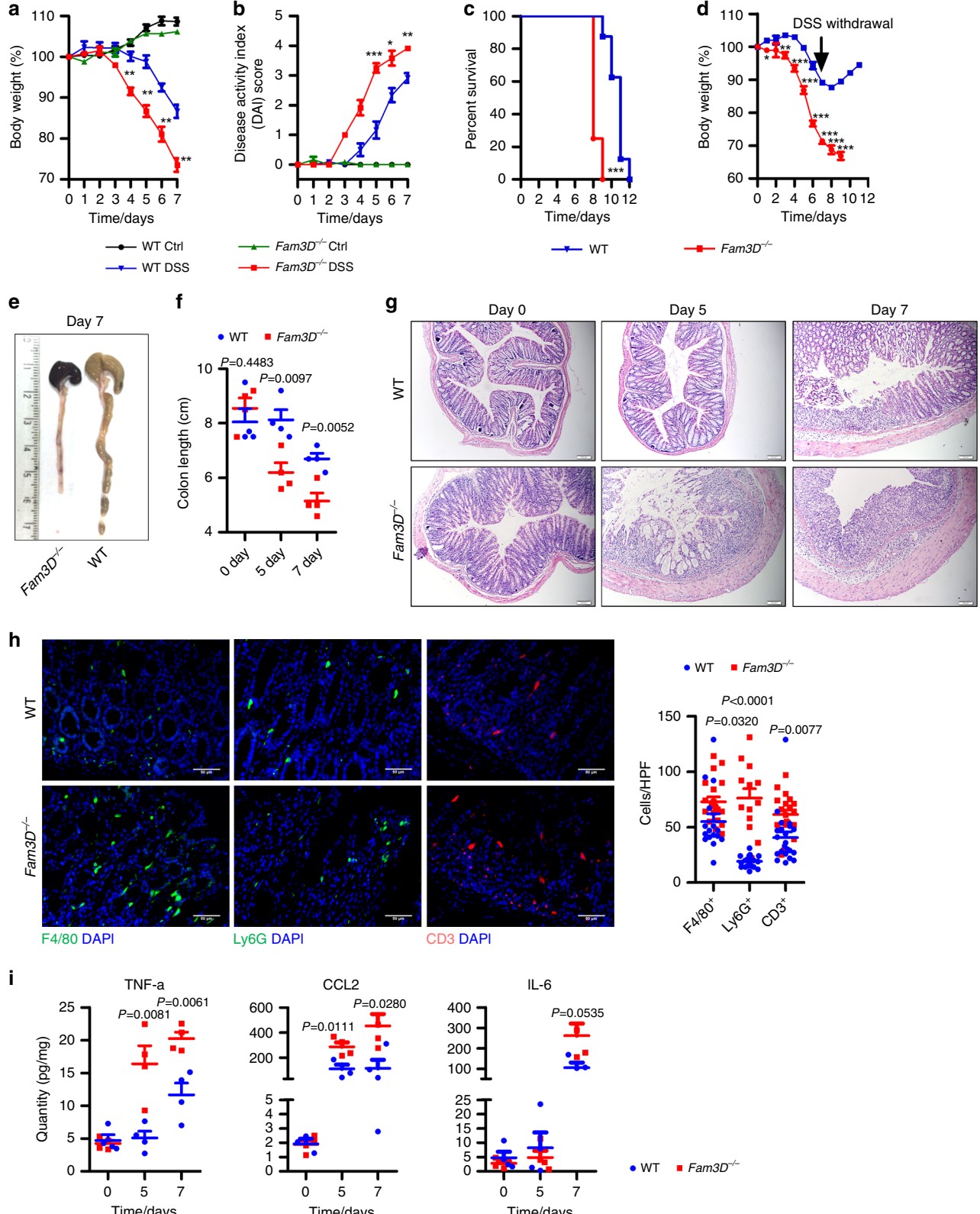

**Fig. 3 Increased severity of DSS-induced colitis in _Fam3D_$^{-/-}$ mice. a** Body weight of WT and _Fam3D_$^{-/-}$ mice given 2.5% DSS in drinking water for 7 days; $n = 4$. **b** Disease scores. **c** The survival of mice. $n = 8$ per group. Mouse death was defined by 35% body weight loss. **d** Body weight. $n = 4$ per group. **e, f** Colon length. $n = 4$ per group. **g** Distal section of the colons examined by H&E staining. Scale bar $= 100\,\mu m$. **h** F4/80-positive macrophages (green, left panel), Ly6G-positive neutrophils (green, middle panel) or CD3-positive lymphocytes (green, right panel) infiltrating the colon of WT and _Fam3D_$^{-/-}$ mice. Scale bar $= 50\,\mu m$. Positive cells in 10 HPF were counted based on six sections from four independent mice. **i** The expression of TNF-α, CCL2 and IL-6 in the colon examined by cytometric bead assays. $n = 4$ per group. Data is representative of one experiment repeated three independent times. Data is presented as the mean ± SEM. Statistical significance was determined by unpaired, two-tailed Student's _t_ test.

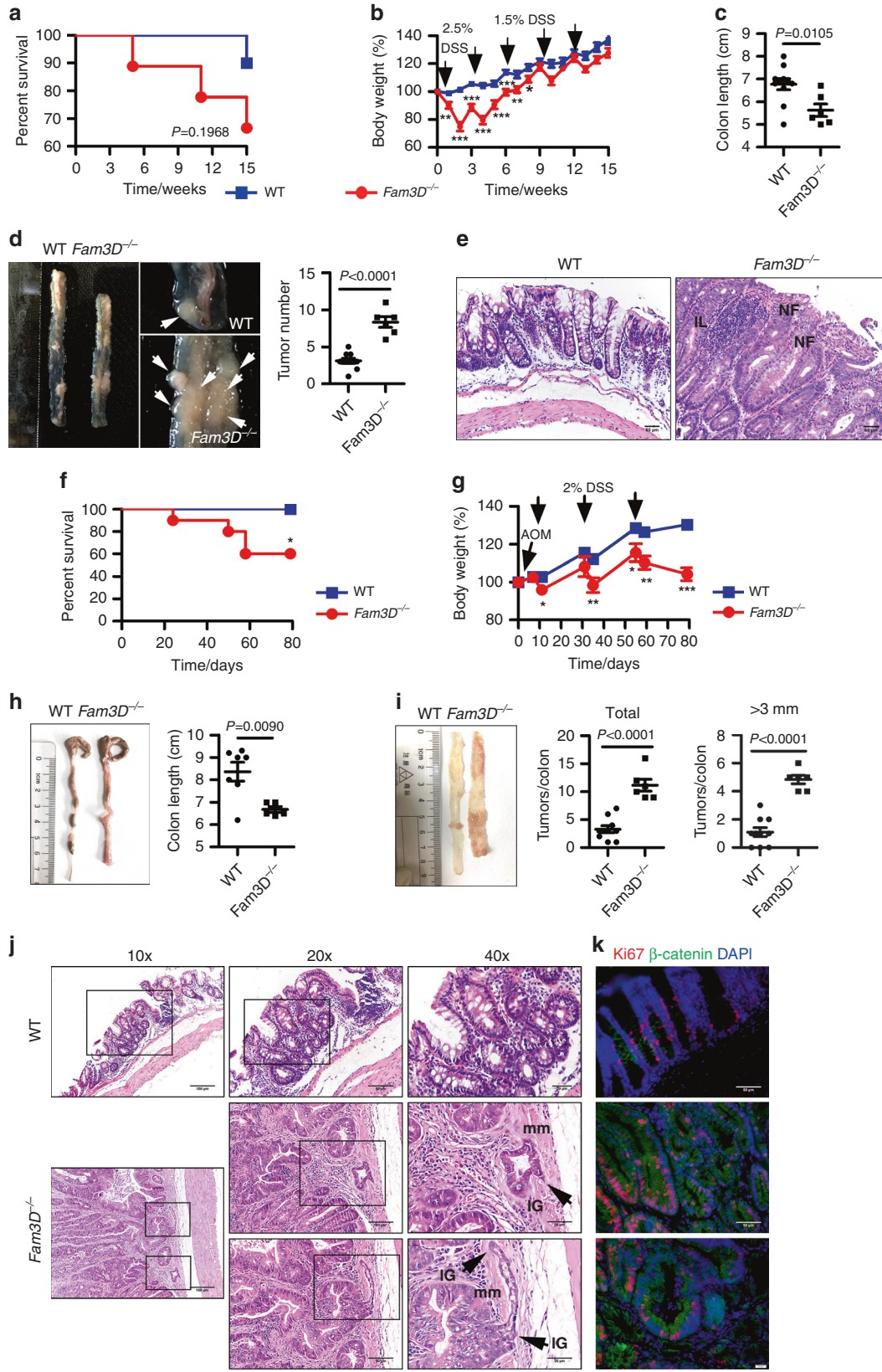

persisted for 5 days after DSS treatment. To obtain additional evidence supporting the role for Fam3D in the maintenance of microbiota in the colon, we single-housed mice derived from mating pairs of male and female $Fam3D^{+/-}$ littermates and collected fecal pellets for microbiota analysis (Supplementary Fig. 10f). As shown in Supplementary Fig. 10g, the difference in

fecal microbiota composition between $Fam3D^{-/-}$ and WT littermates showed divergence after 3-month of single-housing, and the composition of the microbiota in heterozygous littermates remained similar to those of WT littermates. The increased abundance of Bacteroidetes was identified in single-housed $Fam3D^{-/-}$ mice (Supplementary Fig. 10h), consistent

**Fig. 4 Increased severity of DSS-induced chronic colitis and AOM/DSS-induced carcinogenesis in _Fam3D_$^{-/-}$ mice.** WT ($n = 10$) and _Fam3D_$^{-/-}$ mice ($n = 9$) were given 1.5% DSS in drinking water for 4 days followed by normal water for 2 weeks for 5 cycles. On week 15, mice were euthanized and colons were harvested. **a** Survival curves of mice. **b**, **c** Changes in body weight (%) and colon length (cm). **d** The number of tumors in the colon of WT and _Fam3D_$^{-/-}$ mice. White arrow: tumors. **e** Tumors in the colon examined by H&E staining. Scale bar = 50 μm. NF: neoplastic foci; IL: infiltrated leukocytes. Mice were given 12.5 kg g−1 AOM and followed by 1.5% DSS in drinking water for 3 days. Mice were then given normal water for 2 weeks. The treatment was repeated 3 cycles. **f–i** Survival curves of mice, body weight loss, colon length and tumor numbers. **j–k** Tumors developed in the colon of WT and _Fam3D_$^{-/-}$ mice examined by H&E staining (**j**). Scale bar = 100 μm (left panels); Inset scale bar = 50 μm (middle panels) and 20 μm (right panels). IG: invasive gland; mm: muscularis mucosa; asterisk: atypical glandular architecture. Dual immunofluorescent staining of β-catenin (green) and Ki67 (red) in colon tumors of WT and _Fam3D_$^{-/-}$ mice (**k**). Scale bar = 50 μm. Data is representative of one experiment repeated two independent times. Data is presented as the mean ± SEM. Statistical significance was determined by unpaired, two-tailed Student's _t_ test.

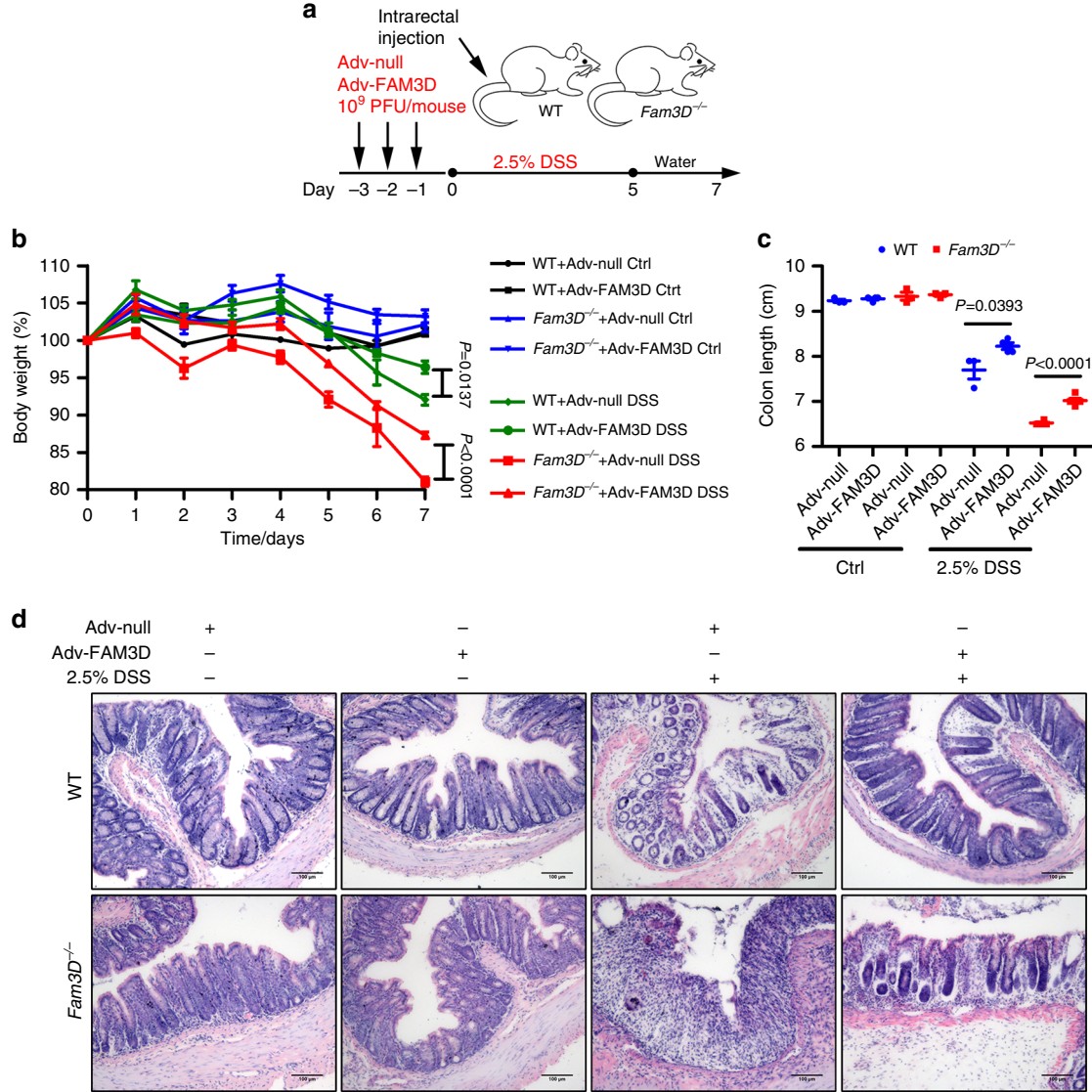

**Fig. 5 Rescue of established colitis by Adv-FAM3D in _Fam3D_$^{-/-}$ mice. a** Protocol for intrarectal infusion of adenovirus. **b, c** Changes in body weight (%) and colon length (cm). **d** Colons examined by H&E staining. Scale bar = 100 μm. Data is representative of 1 experiment repeated 2 independent times. Data is presented as the mean ± SEM. Statistical significance was determined by unpaired, 2-tailed Student's _t_ test.

with earlier experiments (Fig. 6h). These results further support the role of Fam3D in maintaining microbiota balance in the mouse colon.

**Significant reduction in FAM3D expression by human CRC.** We then examined the potential clinical relevance of FAM3D to human colon cancer progression. Analysis of TCGA and CPTAC data revealed that FAM3D mRNA and protein were more highly expressed in normal human colon tissues than in CRC tissues (Supplementary Fig. 11a, b). In addition, while FAM3D transcripts were expressed at lower levels in all stages of CRC (Supplementary Fig. 11c). FAM3D protein was more poorly expressed in higher stages of CRCs, with the lowest level in stage IV cancers (Supplementary Fig. 11d). These results are in accordance with our observations from animal studies, where

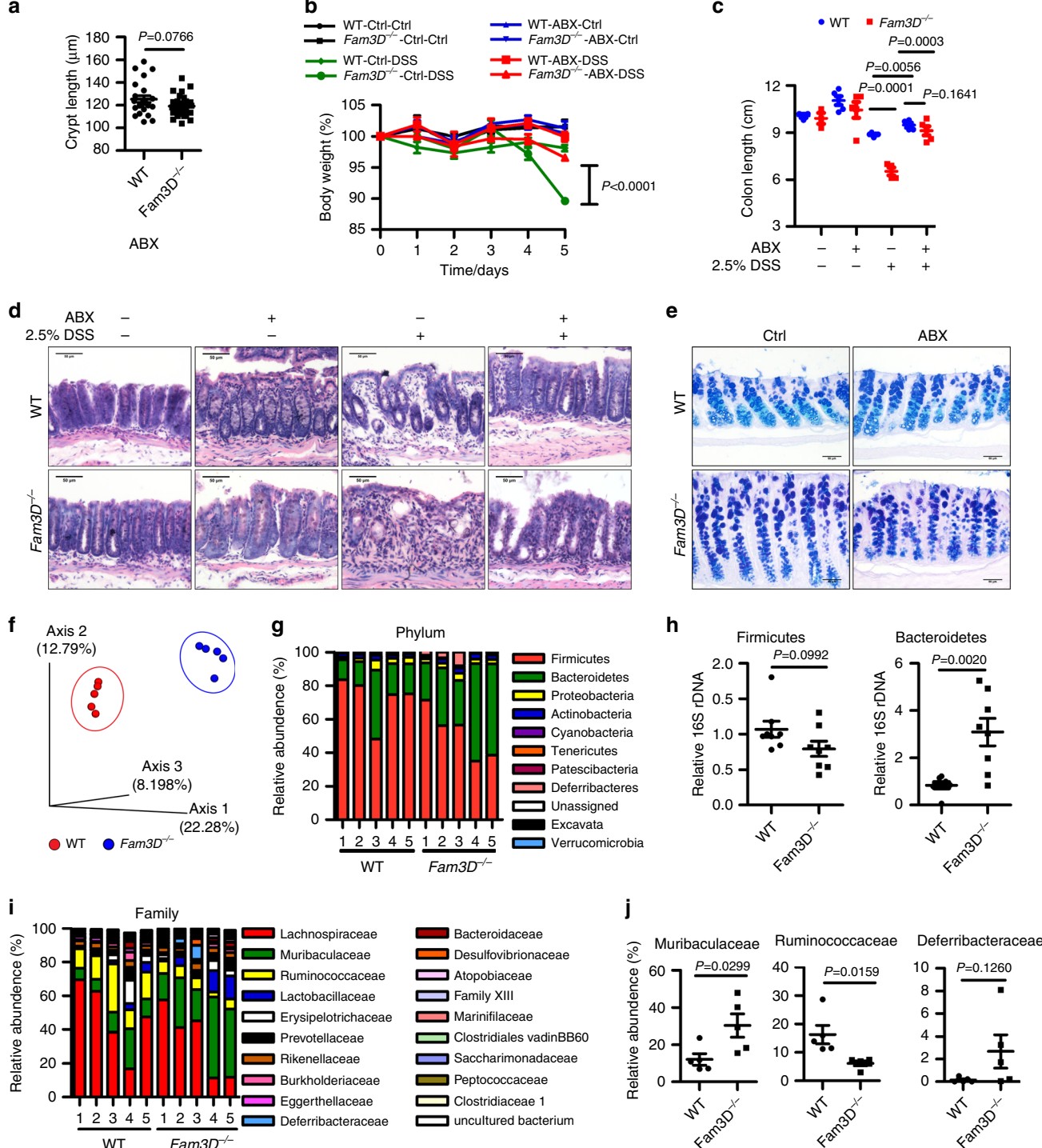

**Fig. 6 Altered composition of microbiota in the colon of *Fam3D* deficient mice.** WT and *Fam3D*<sup>−/−</sup> mice were pretreated with a cocktail of ABX (vancomycin, imipenem/cilastatin, and neomycin) for 4 weeks prior to administration with 2.5% DSS in drinking water. **a** Crypt length of mouse colon treated with ABX. **b, c** Body weight and colon length. **d** Histological changes examined by H&E. Scale bar = 50 μm. **e** Representative Alcian blue staining. Scale bar = 50 μm. Data is representative of 1 experiment repeated 2 independent times. **f** Principal-component analysis of the Jaccard emperor visualized in naïve mice. **g** Analysis of specific bacterial 16S rDNA to detect commensal diversity in WT and *Fam3D*<sup>−/−</sup> mice at the phylum level. **h** Relative abundance of fecal microbiota at the phylum level confirmed by real-time PCR. **i** Analysis of specific bacterial 16S rDNA to detect commensal diversity in WT and *Fam3D*<sup>−/−</sup> mice at the family level. **j** Families selected from mice with given genotypes. For microbiome sequencing, $n = 5$ for each group. For real-time PCR, $n = 8$ for each group. Data is presented as the mean ± SEM. Statistical significance was determined by unpaired, 2-tailed Student's *t* test.

deficiency in *Fam3D* resulted in the disruption of colon homeostasis accompanied by increased carcinogenesis. Therefore, FAM3D (Fam3D) represents an anti-cancer molecule in both human and mice.

## Discussion

To understand how gut homeostasis is maintained and regulated based on the complex interplay between intestinal epithelial cells, the immune system, and the microbiota are areas of intense

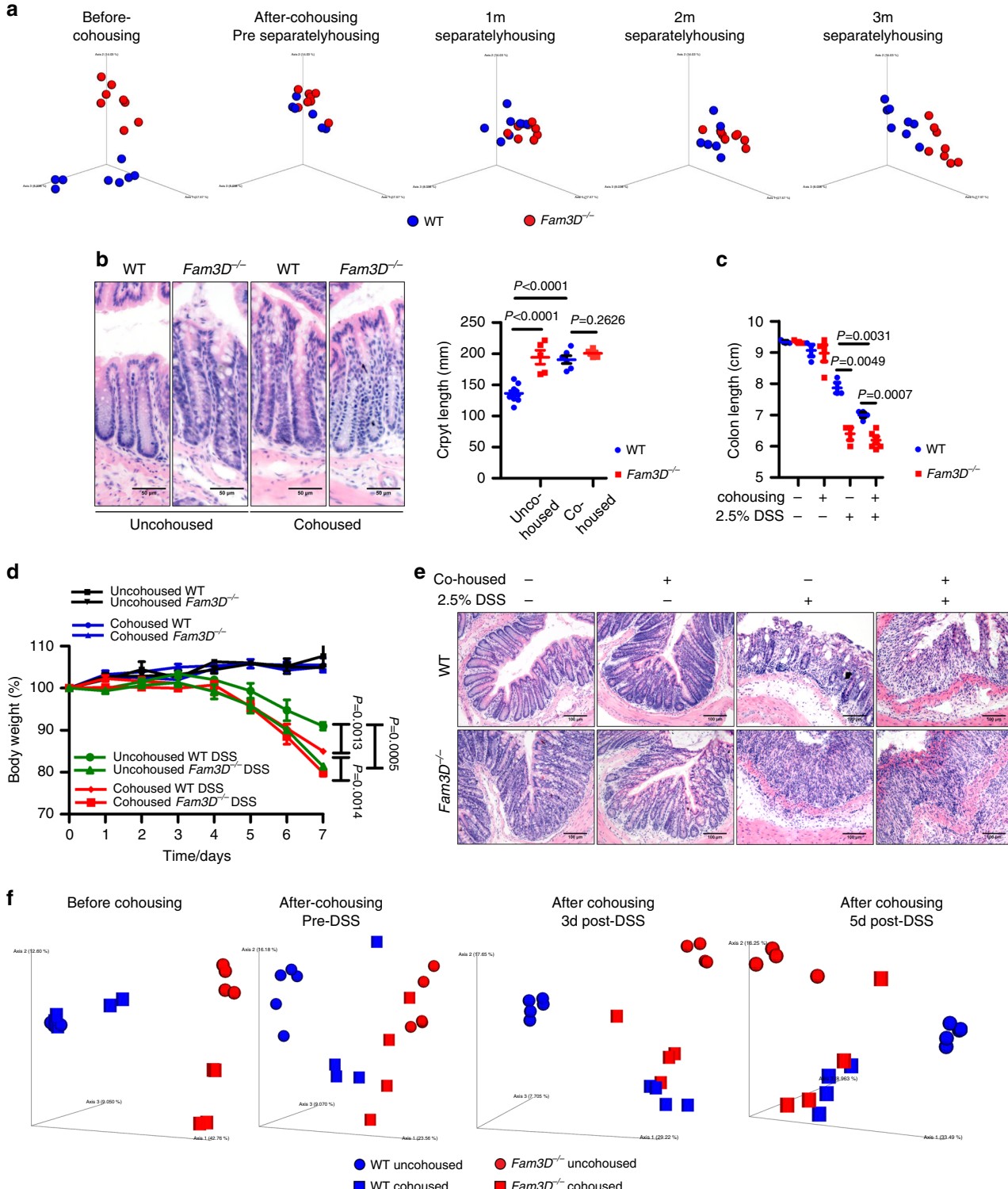

**Fig. 7 Transfer of *Fam3D*⁻/⁻ mouse phenotype to WT mice.** Mice were co-housed for 4 weeks and then separated for 3 months to examine the divergence of the microbiota. **a** Principal-component analysis of Bray-Curtis visualized at different time points. *n* = 8 for each group. **b** Representative crypt length after 4-week cohousing without DSS treatment. **c**, **d** Body weight and colon length on day 7 of DSS treatment. **e** Histological changes examined by H&E. Scale bar = 50 μm. Mice were treated with 2.5% DSS before and after cohousing at 3 and 5 days. Fecal bacterial DNA was isolated from uncohoused and cohoused WT and *Fam3D*⁻/⁻ mice. Data is representative of 1 experiment repeated 2 independent times. **f** Principal-component analysis of the Jaccard emperor visualized at different time points. Data is presented as the mean ± SEM. Statistical significance was determined by unpaired, 2-tailed Student's *t* test.

research. The aim of our current study was to clarify the role of a gut-secreted cytokine-like molecule, FAM3D (mouse Fam3D), in healthy and inflamed colon.

We confirmed that *Fam3D* deficiency impaired mucosal barrier function with reduced expression of acidic mucins, the thickness of the mucus layer and reduced antimicrobial peptides, which were associated with gut dysbiosis. Since mucins with terminal sulfate and sialic acid residues can be cleaved by bacterial sialidases and glycosulfatases, and oligosaccharide side chains are degraded by linkage-specific glycosidases[23], loss of acidic mucins may cause alterations in microbiota habitat and further impair the normal bacterial composition. Antimicrobial peptides, including REG family and defensins, participate in the patho-physiological processes in the colon by maintaining the homeostasis of microbiota[7–9]. There is a close link between microbiota and the immune system in the gastrointestinal tract. Chemically or pathogen-induced inflammation in the gut disrupts the homeostasis of the microbiota and promotes the development of inflammatory diseases including IBD[21], since dysbiosis is prone to potently activate pattern recognition receptors such as Toll-like receptors and Nod-like receptors on both epithelial and resident macrophages to release chemoattractants, in addition to those produced by the bacteria per se[10,24] to enhance the recruitment and activation of additional immune cells to exacerbate inflammatory responses. The spontaneous and DSS-induced colitis shown in *Fam3D*$^{-/-}$ mice bears similarities to those reported in *O-glycans*, *Muc2*, *Cramp*, and *T-bet*-deficient mice[25–27], which all showed dysbiosis in association with aberrant inflammatory responses culminating in increased carcinogenesis.

There were similarities in the phenotype among *Fam3D*$^{-/-}$, *Fpr2*$^{-/-}$, and *Fpr1/2*$^{-/-}$ mice, and mice deficient in another Fpr2 agonist *Cramp* in that they all showed increased susceptibility to chemically induced colitis. FPRs (mouse Fprs) are members of G protein-coupled chemoattractant receptors with 7-transmembrane structure and were originally identified as regulators of the recruitment of phagocytic leukocytes and activation of microbicidal respiratory burst[28]. It has been shown that Fpr2 recognizes a variety of pathogen and host-derived chemotactic agonists[29]. Functional FPRs are also identified in IECs that serve as receptors for a variety of pathogen- and host-rederived ligands that regulate mucosal homeostasis[30–33]. In *Fpr2*$^{-/-}$ mice, colonic epithelial cells display defects in commensal bacterial-dependent homeostasis, including shortened colonic crypts, reduced acute inflammatory responses, and delayed mucosal restoration after injury[33]. CRAMP is an antibacterial peptide and participates in important patho-physiological responses by maintaining the balance of microbiota in the colon[34]. *Cramp* deficient mice phenocopied *Fpr2*$^{-/-}$ mice showing shortened colonic crypts and were highly sensitive to chemically induced colitis and carcinogenesis[34]. However, elongated crypts were observed only in *Fam3D*$^{-/-}$ mice, suggesting a unique role of Fam3D in regulating the growth of crypts, which may be mediated through the microbiome, because *Fam3D*$^{-/-}$ mice treated with antibiotics show restoration of the colon crypt length. Nevertheless, further study is required to clarify the contribution of Fprs expressed by colon epithelial and myeloid cells to the phenotype of *Fam3D* deficiency.

It is interesting to note that *Fam3D*$^{-/-}$ mice continued to show a mild degree of colitis with mucosal damage and lower degree of lymphocytic infiltration after elimination of microbiota by antibiotics, suggesting dysbiosis may not fully explain the role of Fam3D in intestinal homeostasis and inflammation. This mechanistic segregation from microbiota-dependent phenotype needs further investigation. By comparison between WT and *Fam3D*$^{-/-}$ mice, we found there was a goblet cell abnormality in *Fam3D*$^{-/-}$ mouse colon, with increased number and size, but

PAS-positive cells showing only in lower crypts in the absence of acidic mucins. These observations suggest impaired secretory functions of goblet cells[35,36]. We noticed that in the colonic epithelium in *Fam3D*$^{-/-}$ mice at different mouse ages, goblet cell dysfunction precedes the elongation of crypts and these abnormalities result in mucus retention over time, which in turn contributes to progressive dysfunction of the intestinal barrier. Therefore, we propose that alteration of colonic mucosa acidic mucins is an intrinsic defect caused by *Fam3D* deficiency that may contribute to the weakening of mucosal barrier function associated with dysbiosis arising from reduced production of antimicrobial components in the colon.

Studies of two *Muc2* mutants (*Winnie* and *Eeyore*) demonstrate that morphological changes in goblet cells, including misfolding/aberrant assembly of MUC2 precursor causes accumulation of immature MUC2, resulting in substantial endoplasmic reticulum stress that contributes to epithelial damage and chronic inflammation[35]. Increased number and size of goblet cells but reduction in secreted acidic mucins in the colon of *Fam3D*$^{-/-}$ mice suggest that Fam3D may participate in the regulation of mucin secretion and/or stability. Further study is ongoing to clarify the nature of cellular receptors or sensors for Fam3D to exert its critical function in the colon.

As intestinal homeostasis provides the habitant environment to commensal bacteria, it is plausible that the dysbiosis seen in *Fam3D*$^{-/-}$ is associated with pathophysiological changes in the colon. These changes include the expansion of potential pathogens including *Muribaculaceae* and *Deferribacteraceae*, and shrinking of beneficial commensal species i.e *Ruminococeaceae*, as an important short-chain fatty acid producer[37], and Segmented filamentous bacteria, as an antibacterial peptide inducer[38]. This appears to be consistent with several mouse models that exhibit barrier defects such as *Muc2* and *Nod2* deficient mice[39,40]. Additionally, a large proportion of bacterial species that were increased relative abundance in *Fam3D*$^{-/-}$ gut are also related to mucus degradation thus considered as colitogenic including *Peptostreptococcus* and *Clostridium perfringens*. *Peptostreptococcus*, more abundant in *Fam3D*$^{-/-}$ mice, is also significantly enriched in patients with UC and CRC[41]. In addition, *Clostridium perfringens* exerts proteolytic and mucinase activity, both of which may play a role in the pathogenesis of pouchitis, a colon inflammatory disease[42] and may be attributed in part to a thinner mucus layer observed in the colon of *Fam3D*$^{-/-}$ mice. In contrast, *Lactobacillus/Lactococcus* and *Enterococcus faecalis* (*E. faecalis*) showed unpredictable changes in the colon of *Fam3D*$^{-/-}$ mice. Some *Lactobacilli/Lactococcus* strains are potential probiotics as they can maintain gut homeostasis. *E. faecalis*, as a proinflammatory bacterium, which is found to correlate strongly with murine colitis, may have been involved in the development of IBD, with yet to be confirmed relevance to human IBD[43,44]. *E. faecalis*, as a proinflammatory bacterium, is found to correlate strongly with IBD status[45]. However, similar intestinal bacterial pattern has been described in *Nod2*$^{-/-}$ mice with the elimination of anaerobic bacteria[40], indicating abnormally increased *Lactobacillus* might be a result of the imbalanced composition of commensal microbiota and also as a sign of dysbiosis. These results highlight the complexity of microbiota that undergoes dynamic changes with the microenvironment and certain species should be isolated and future studies using complementation of dysbiosis microbiome are warranted to decipher the exact significance of the changes in microbiota composition shown in *Fam3D*$^{-/-}$ mice.

Our study demonstrated an important role of Fam3D in supporting the development of the normal architecture of mouse colon in which the absence of the molecule is associated with

abnormal development of the colon crypts, increased spontaneous low-level inflammation, reduced secretory function of goblet cells causing thinner layer of protective mucins on the mucosal surface as well as a reduction in key anti-microbial molecules, culminating in dysbiosis, which exacerbates the colitis with markedly increased incidence of cancer. These findings may have clinical relevance since the human counterpart protein FAM3D was more highly expressed in normal colon tissues in contrast to CRC tissues with the lowest levels of both FAM3D mRNA and protein expressing in Grade IV CRC, with which the patients suffer from the poorest survival. Thus, the impact of Fam3D on colon homeostasis and diseases is multifaceted and its capacity to protect normal colon should be exploited as a future therapeutic modality, as revealed in our study that anal administration of adenovirus-encoded FAM3D significantly improved the severity of colitis in mice. It is therefore promising that FAM3D may be administered in human for the management of colitis and associated cancer by using safer and reliable recombinant bacterial delivery systems. However, DSS-induced colitis in mice has limited representation to human IBD, therefore more suitable models such as T-cell-transferred chronic colitis should be considered in future studies to clarify the significance of FAM3D in the pathogenesis of human IBD and associated cancer as well as a therapeutic agent.

In conclusion, FAM3D (Fam3D) is identified as a protector of the mouse colon. Its capacity to sustain the normal differentiation of goblet cells, their mucin production, and the presence of anti-microbial molecules is critical for a balanced microbiome in the gut. Therefore, FAM3D (Fam3D) represents a class of guardians of the gut (Supplementary Fig. 12).

## Methods

**Mice.** Fam3D[−/−] mice were generated on a C57BL/6 background[15] and the genotype was confirmed by genomic DNA sequencing (Genewiz, South Plainfield, NJ). Wild type (WT) mice were produced from Fam3D[+/−] mice and maintained in specific-pathogen-free facility at the Health Science Center, Peking University, and National Cancer Institute at Frederick, Frederick, MD. The experimental protocols of this study were approved by the Ethics Committee of Peking University Health Science Center (LA2016010) and the Frederick National Laboratory for Cancer Research Animal Care and Use Committee, Frederick, MD. All experiments were performed in accordance with procedures outlined in the "Guide for Care and Use of Laboratory Animals" (National Resource Council, National Academy Press, Washington D.C).

**Antibodies and other reagents.** Goat anti-mouse FAM3D antibody (AF3027) and goat anti-human FAM3D antibody (AF2869) were purchased from R&D Systems (Minneapolis, MN). Rabbit anti-β-actin (4970) and glyceraldehyde-3-phosphate dehydrogenase (GAPDH) (5174) were purchased from Cell Signaling Technology (Boston, MA). 4′,6-diamidino-2-phenylindole (DAPI) (Millipore Sigma, MA) used to stain cell nuclei and fluorescein isothiocyanate (FITC) conjugated Ulex europaeus agglutinin-I (UEA-I) were obtained from Sigma-Aldrich (MA, US). Rat anti-CD3 (ab11089), rabbit anti-B220 (ab64100), rabbit anti-F4/80 (ab111101), Rat anti-Ly6G (ab25377), rabbit anti-Ki67 (ab16667), rabbit anti-Chromogranin A (CHGA) (ab15160), goat-anti rabbit IgG horseradish peroxidase (HRP)-conjugated (ab6721) and rabbit anti-goat IgG HRP-conjugated (ab6741) antibodies were purchased from Abcam (Cambridge, UK). Rat anti-EpCAM antibody (sc-53532), mouse anti-β-catenin antibody (sc-7963), rabbit anti-mouse Mucin 2 (MUC2) antibody (H300) were purchased from Santa Cruz (Dallas, TX). Rabbit anti-mouse Reg3γ (PA5-50450), goat anti-rabbit IgG Alexa Fluor 488 and 568 (A-32731 and A-11011), donkey anti-rat IgG Alexa Fluor 488 and 594 (A-21208 and SA5-10028) and rabbit anti-goat IgG Alexa Fluor 568 (A-11079) antibodies were purchased from Invitrogen (San Diego, CA). Antibiotics were purchased from the following companies: vancomycin (Hospira, Lake Forest, IL), imipenem/cilastatin (brand name Primaxin, Merck, Bethesda, MD), and neomycin (Med-Pharmex, Pomona, CA).

**Cell lines and culture conditions.** 293 T cells (ATCC) were cultured in high-glucose (4.5 g/L) Dulbecco's modified Eagle's medium (DMEM, Gibco, San Diego, CA) supplemented with 10% fetal bovine serum (FBS, Hyclone, Logan, UT), 100 U/mL penicillin, 100 μg/mL streptomycin (Invitrogen Life Technologies), 15 mM 4-(2-hydroxyethyl)-1-piperazineethane sulfonic acid (HEPES) (pH 7.4, Invitrogen Life Technologies, San Diego, CA). Human epithelial cell line FPCK-1-1 was grown in DMEM (Gibco, San Diego, CA) with 10% FBS, 100 U/mL penicillin, 100 μg/mL streptomycin and 15 mM HEPES.

**The expression and purification of recombinant mouse Fam3D protein.** The plasmid pcDNA3.1-Fam3D-myc-his was transfected with polyethyleneimine (PT-101, Polyplus, Berkeley, CA) into 293 T cells. After 16 hour (h), 293 T culture medium was replaced with Hektor S medium (Cell Culture Technologies, Switzerland) with 2% glutamate (Sigma-Aldrich). The supernatant of transfected 293 T cells was collected at 6 days post transfection and Fam3D protein was purified by Ni-Sepharose High-Performance system (GE Healthcare)[14].

**Western blotting.** Tissues or cells were lysed in RIPA buffer (Invitrogen) containing protease and phosphatase inhibitors (Sigma-Aldrich). Equal amounts of total protein were loaded on and separated by 4%-12% sodium dodecyl sulfate (SDS)-PAGE (Invitrogen) and then transferred onto polyvinylidene difluoride membranes (Millipore, Billerica, MA). The membranes were blocked for 1 h in Tris-buffered saline containing 5% bovine serum albumin (BSA) (w/v) and 0.1% Tween-20 (v/v) and incubated with indicated primary antibodies overnight at 4 °C, followed by incubation for 1 h with appropriate secondary antibodies. Protein bands were detected with a Super Signal Chemiluminescent Substrate (Pierce, Rockford, IL) and developed using a G: Box gel doc system (Syngene, Frederick, MD). Densitometer analysis was performed using the ImageJ analysis program (National Institutes of Health).

**Reverse transcription and real-time PCR.** Fresh mouse tissues or cells were submerged in RNAlater (Thermo Fisher Scientific, Waltham, MA) at 4 °C overnight. Total RNA was extracted and purified by using the RNeasy Mini Kit (Qiagen, Germantown, MD) according to the manufacturer's protocol. Equal amounts (2 μg) of total RNA were reverse-transcribed to cDNA by using All-in-One RT MasterMix (Applied Biological Materials Inc., Richmond, Canada). After reverse transcription, real-time PCR amplification was performed using SYBR Green qPCR Mastermixes (Qiagen, Germantown, MD) under a 7500 Real-Time PCR System (Applied Biosystems, NY). PCR program consisted of 95 °C for 5 min (min); 40 cycles of 95 °C for 30 s (s), 55 °C for 30 s and 72 °C for 30 s; and 72 °C for 5 min. The expression of candidate genes in each group of mice was normalized to Hprt to obtain a ΔCt value and to calculate a 2^-(mean ΔCt). Sequences of the primer used for PCR amplification are listed in Supplementary Table 1.

**Histological analyses and immunohistochemistry.** For histological analysis, mouse colons were fixed in neutral buffered formalin or in Carnoy's fixative (60% methanol, 30% chloroform, and 10% acetic acid) and embedded in paraffin. Tissue sections 5 μm were stained with hematoxylin and eosin. For glycan detection, Periodic acid-Schiff (PAS) (Sigma-Aldrich) and Alcian blue (Sigma-Aldrich) were used to stain general intestinal carbohydrate moieties; high iron diamine (HID) solution (120 mg N, N-dimethyl-m-phenylenediamine dihydrochloride, 20 mg N, N-dimethyl-p-phenylenediamine monohydrochloride, and 1.4 mL 10% FeCl$_3$ in 50 mL H$_2$O) (Sigma-Aldrich) was used to stain acidic mucins.

For immunohistochemistry, deparaffinized tissue sections were subjected to antigens rederivation, peroxidase removal, 3% BSA blockage, and then were incubated with indicated primary antibodies at 4 °C overnight followed by appropriate secondary HRP-conjugated antibodies before staining with 3,3′-diaminobenzidine Substrate Kit (Pierce, Rockford, IL). Nuclei were counterstained with hematoxylin. Sections incubated with species-matched IgG alone were used as negative controls.

For immunofluorescence, deparaffinized tissue sections underwent 5% BSA blockage, then were incubated with indicated primary antibodies at 4 °C overnight followed by incubation with appropriate secondary fluorescent-conjugated antibodies and DAPI. Coverslips were mounted in Fluorescence Mounting Medium (Dako, Glostrup, Denmark) for visualization.

**FISH and dual UEA-I/FISH labeling.** For Fluorescence in situ hybridization (FISH), colons containing stool were fixed with Carnoy's solution and embedded in paraffin. Deparaffined and rehydrated sections were incubated with Cy3-conjugated universal bacterial probe EUB338 (5′-GCTGCCTCCCGTAGGAGT-3′) or with a nonspecific probe NON338 (5′-ACTCCTACGGGAGGCAGC-3′) as a negative control (synthesized by Integrated Device Technology, Inc., San Jose, CA) in hybridization buffer (20 mM Tris-HCl, pH 7.4; 0.9 M NaCl; and 0.1% SDS) at 42 °C overnight. The sections were submerged in buffer (20 mM Tris-HCl, pH 7.4; 0.9 M NaCl) at 42 °C for 15 min. For dual UEA-I/FISH labeling, sections were stained with FITC-labeled UEA-I at room temperature for 2 h after FISH[46].

**TUNEL assay.** Terminal deoxynucleotidyl transferase dUTP nick end labeling (TUNEL) assay was performed with the TUNEL Apoptosis Detection Kit (Alexa Fluor 488) (YEASEN, Shanghai, China) according to the manufacturer's recommendations. Briefly, rehydrated sections were incubated with 20 μg/mL Proteinase K at 37 °C for 20 min followed by 1×Equilibration Buffer at room temperature for 30 min and then incubated with DNA Labeling Solution (For 50 μL DNA Labeling Solution: 10 μL 5×Equilibration Buffer + 5 μL Alexa Fluor 488-12-dUTP Labeling

Mix + Recombinant TdT Enzyme + 34 μL H$_2$O) at 37 °C for 60 min in the dark. The slides were rinsed with Wash Buffer (0.1% Triton X-100 + 5 mg/mL BSA in phosphate-buffered saline (PBS)) for three times followed by 2 μg/mL DAPI at room temperature for 5 min in the dark. Coverslips were mounted for visualization.

**Isolation of colonic epithelial cells and lamina propria cells.** Colons removed of fecal contents were opened longitudinally and cut into 1 cm pieces in ice-cold PBS. The pieces of the intestine in 5 mL of colonic epithelial cells (CECs) digestion solution (5 mmol/L ethylenediaminetetraacetic acid and 1 mmol/L dithiothreitol in Ca$^{2+}$/Mg$^{2+}$-free Hank's Balanced Salt Solution) were placed on an orbital shaker for 20 min at 37 °C at 40 gravity (×g). Cell suspension obtained was passed through a 100 μ μm cell and centrifuged to obtain CECs. The remaining tissues were washed by PBS and cut into 1 mm pieces, then incubated in 5 mL of cLP (colonic lamina propria) digestion solution (0.5 g/L collagenase D, 0.5 g/L DNase I in PBS) at 37 °C for 20 min under slow rotation. After incubation, the cell solution was vortexed rigorously for 20 s and filtered through a 40 μm strainer. The supernatant was collected and centrifuged for 10 min at 500 × g at 20 °C. The cell pellet was washed in cold fluorescence-activated cell sorting (FACS) buffer (3% FBS (v/v) in PBS) and centrifuged at 500 × g for 10 min at 20 °C[47]. The pellet was resuspended in cold FACS buffer for evaluation.

**Flow cytometry.** Fluorescence-labeled anti-mouse CD45-FITC (103108), anti-mouse CD11b-APC-Cy7 (101226), anti-mouse F4/80-PE (123109), anti-mouse Ly6G-PerCP-Cy5.5 (127616),anti-mouse Ly6C-APC (128016), anti-mouse CD3-PE (100206), anti-mouse B220-APC (103212) antibodies were obtained from BioLegend (San Diego, CA). All antibodies were used at 1:100 dilutions. For analysis of leukocytes from euthanized mice, peripheral blood cells were collected through cardiac puncture, followed by elimination of red blood cells by using a lysis buffer (Tiangen Biotech Inc, Beijing, China). The cells were also isolated from the colons as described above. cLP cells were incubated in FACS buffer with indicated antibodies for 30 min at 4 °C in the dark. After the removal of unbound antibodies, BD FACSVerse™ System (BD Biosciences, San Diego, CA) and FlowJo software were used for data acquisition and analysis.

**Isolation and culture of colonic crypts.** Colonic fragments were washed with cold PBS until the supernatant was clear, then tissue fragments were incubated in Gentle Cell Dissociation Reagent (STEMCELL Technologies, Seattle, WA) at room temperature on a rocking platform at 20 rpm for 20 min. The tissue pieces were sedimented by gravity for approximately 30 s and resuspended in cold PBS with 0.1% BSA (w/v) and pipetted 3 times. Colonic pieces settled to the bottom were filtered through a 70 μm strainer. The procedure was typically repeated 6–8 times. The supernatants containing crypts were then collected, washed with cold chelation buffer, and centrifuged at 290 × g for 3 min. The pellets were resuspended in cold PBS with 0.1% BSA (w/v), transferred into a new tube, then centrifuged at 200 × g for 3 min at 4 °C. Each crypt pellet was resuspended in cold DMEM/F-12 with 15 mM HEPES (STEMCELL Technologies, Seattle, WA) and centrifuged at 200 × g for 3 min at 4 °C for final collection[48].

Isolated colonic crypts were counted using a hemocytometer and then were embedded in Matrigel on ice (growth factor reduced, phenol red free; BD Biosciences) and seeded in 48-well plates (500 crypts in 50 μL of Matrigel per well). The Matrigel was polymerized for 10 min at 37 °C and 300 μL of complete IntestiCult™ Organoid Growth Medium (STEMCELL Technologies, Seattle, WA) at room temperature were then added to each well. The culture medium was replaced 3 times per week with 300 μL fresh complete IntestiCult™ Organoid Growth Medium at room temperature. For differentiation of the crypts, established organoids were incubated in a complete medium containing 5 μM N-[(3,5-difluorophenyl)acetyl]-L-alanyl-2-phenyl]glycine-1,1-dimethylethyl ester (DAPT, Tocris, Minneapolis, MN), a Notch pathway inhibitor, for 4 days to induce the differentiation of Lgr5$^+$ stem cells[49].

**Analysis of colonic organoids.** Colonic organoids were passaged every 7 days, with an average split ratio of 1:2. Gentle Cell Dissociation Reagent was added on the top of the exposed dome in each well and incubated at room temperature for 1 min. The dome was then broken by pipetting. The suspensions were collected and incubated on a rocking platform at 20 rpm at room temperature for 10 min. The tubes were centrifuged at 290 × g for 5 min at 4 °C, followed by washing with cold DMEM/F-12 containing 15 mM HEPES and then centrifuged at 200 × g for 5 min at 4 °C.

For image analysis, the culture medium was removed, and samples were washed with PBS. Organoids cultured in Matrigel were fixed by directly adding 4% Paraformaldehyde for 30 min at room temperature. Matrigel was then mechanically disrupted and the organoids were transferred into BSA-coated Eppendorf tubes. Samples were washed with PBS, permeabilized with 0.25% Triton X-100 for 30 min, blocked using 3% BSA, and stained with primary and secondary antibodies. Images of organoids were taken by an inverted confocal microscope (Leica SP5, Leica Microsystems Inc, IL).

**RNA-sequencing experiments.** RNA was stored and extracted from Trizol™ and prepared for sequencing by the Mega Genomics Company (Beijing, China). Reads were acquired on an Illumina HiSeq 2000 and aligned to the GRCm38 mouse genome assembly using TopHat2 and counted using HTSeq. Differential expression gene analysis was performed using edgeR. Data were plotted using R. Genes significantly differentially expressed (> 1 log$_2$ fold-change (FC) and < −1 log$_2$ FC, adjusted $P$ value = 0.05). Gene set enrichment analysis (GSEA) was performed (http://www.broadinstitute.org/gsea/index.jsp) on Fam3D$^{-/-}$ vs WT EBSeq log$_2$ FC expression data of colonic epithelial samples. Gene sets comprise stem cell, transit-amplifying cell, enterocytes, goblet cell, Paneth cell, tuft cell, and enteroendocrine cell. Signature genes that each type of intestinal epithelial cells were defined as those significantly enriched as in Yan et al.[16].

**Induction of acute and chronic colitis in mice.** Mice at 6- to 8-week-old were given 2.5% dextran-sulfate-sodium (DSS, w/v, 4000 kDa, MP Biomedicals) in drinking water for 5–7 days. Mice were allowed to recover on normal drinking water for an additional 2 to 5 days. Mice were euthanized with the loss 35% body weight or when they developed a prolapse. To induce chronic colitis, mice were given 1.5% DSS in drinking water for 3 days and followed by normal drinking water for an additional 15 days with 5 cycles of treatment[50]. The disease activity index was calculated as previously described (Supplementary Table 2)[50]. Mice were weighed during the colitis model and sacrificed at indicated time points. Colon tissues were collected for further analyses.

**Induction of colitis-associated carcinogenesis.** Mice were given a single intra-peritoneal injection of azoxymethane (AOM, 12.5 mg kg^−1 body weight; Sigma-Aldrich)[51]. Starting on 7 days after the injection, mice were given 1.5% DSS in drinking water for 4 days and followed by 2 weeks of normal drinking water for 3 cycles. Mice were then sacrificed for histopathological analysis. The colon was cut open longitudinally and examined for the presence of tumors. Tumor numbers were counted and the tumor size (average diameter) was measured with aid of a caliper. Tumor load per mouse was calculated by adding the average diameter of all tumors per colon[51]. Tissues were fixed in 4% formalin and embedded in paraffin for histopathological analysis.

**Antibiotic treatment and co-housing and single-housing of mice.** Mice were given a cocktail of antibiotics (ABX, 500 mg/L vancomycin, 500 mg/L imipenem/cilastatin, and 1 g/L neomycin) in normal drinking water every other day for three weeks[52]. Mice were then given 2.5% DSS dissolved in ABX-containing drinking water for 5 days and followed by ABX-containing drinking water without DSS for 2 days. For co-housing of mice, two WT and two Fam3D$^{-/-}$ mice with same sex were co-housed in a single cage at the time of weaning (4-week of age) and maintained for 4-week prior to use. At least 3 cages were used for co-housing of WT and Fam3D$^{-/-}$ mice. For single-housing, WT, Fam3D$^{+/-}$ and Fam3D$^{-/-}$ littermates were generated by mating paired Fam3D$^{+/-}$ heterozygous mice. Littermates were separated individually at the time of weaning (4-week of age) and maintained for 3 months prior to fecal collection and microbiome sequencing.

**DNA extraction and sequencing of fecal bacteria.** DNA extraction and amplification were performed using Eppendorf liquid handling robots. The V4 region of the 16S rDNA gene (515F-806R) was sequenced for samples; generating paired-end, overlapping reads on the Illumina MiSeq platform[53]. The demultiplexed paired-end fastq files were pre-processed and analyzed using QIIME2 version 2-2018.2 (https://qiime2.org). The DADA2 algorithm[54], implemented in QIIME2, was used for error modeling and filtering the raw fastq files. Post denoising and chimera removal.

Taxonomic classification was performed using the QIIME2 feature-classifier (https://github.com/qiime2/q2-feature-classifier) plugin trained on the Silva 132 database[55]. The 3D PCoA plots were generated using Emperor[56].

**Analysis of human CRC cohorts from UALCAN portal analysis of TCGA and CPTAC dataset.** The expression of FAM3D transcripts and protein was analyzed using data obtained from the Cancer Genome Atlas (TCGA) and Clinical Proteomic Tumor Analysis Consortium (CPTAC) Confirmatory/Discovery dataset[57–60]. Transcripts Per Million (TPM) was a normalization method for RNA-seq. The sum of all normalized transcript expression values was divided by 1,000,000 to create a scaling factor. The normalized expression of each transcript was divided by the scaling factor, which results in the TPM value. TPM values were compared to estimate the relative abundance of the transcripts for cells that have approximately the same number of transcripts per-cell (http:ualcan.path.uab.edu)[57]. RNA-sequencing transcripts were normalized and presented as TPM (http://www.proteinatlas.org)[61]. Z-values represent standard deviation from the median across samples for a given cancer type. Log2 Spectral count ratio values from CPTAC were first normalized within each sample profile, then normalized across samples (http:ualcan.path.uab.edu)[58]. For TCM, $t$ test was performed using a PERL script with Comprehensive Perl Archive Network (CPAN) module "Statistics::TTest". For protein expression, statistical tests were performed using log2-transformed expression values (http:ualcan.path.uab.edu).

**Statistics**. Unless otherwise specified, statistical analysis was performed by using Student's *t*-test or Log-rank (Mantel-Cox) test and analyzed using GraphPad Prism V5.0 (San Jose, CA). Data are presented as the mean ± standard error of the mean (SEM). A value of $P < 0.05$ was considered statistically significant.

**Reporting Summary**. Further information on research design is available in the Nature Research Reporting Summary linked to this article.

## Data availability

Source data are provided with this paper. GSEA was performed (http://www.broadinstitute.org/gsea/index.jsp) on *Fam3D*⁻/⁻ vs WT EBSeq log₂ FC expression data of colonic epithelial samples. RNA-sequencing data is deposited on the Short Read Archive (SRA) of NCBI: BioProject PRJNA641502. 16S sequencing data are deposited on the SRA of NCBI: PRJNA638922, PRJNA638979, and PRJNA638615. In accordance with the data access policies of the TCGA and CPTAC projects, most molecular, clinical and specimen data are in an open tier that does not require access approval. The expression of FAM3D transcripts and protein shown in Supplementary Fig. 11 was based upon the information contained in TCGA and CPTAC datasets analyzed by UALAN portal (http://ualcan.path.uab.edu/cgi-bin/TCGAExResultNew2.pl?genenam=FAM3D&ctype=COAD, for Supplementary Fig. 11a, b; http://ualcan.path.uab.edu/cgi-bin/CPTAC-Result.pl?genenam=FAM3D&ctype=Colon, for Supplementary Fig. 11c, d). Source data are provided with this paper.

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

## Acknowledgements

We are grateful to Professor Dalong Ma (Department of Immunology, School of Basic Medical Sciences, NHC Key Laboratory of Medical Immunology and Center for Human Disease Genomics, Peking University) for the omics strategies and valuable suggestions. We thank Dr. A. Tominaga of Kochi University, Kochi, Japan for kindly providing human colon cell line FPCK-1-1. The authors also thank Dr. J. J. Oppenheim for reviewing the manuscript and Dr. Dennis Klinman for providing intellectual input. The technical assistance provided by CIP Mouse Core and by Mr. Timothy Back, NCI, the staff of LASP, Leidos, is gratefully acknowledged. The secretarial assistance by Ms. Cheri A. Rhoderick is also appreciated. This project has been supported in part by the National Natural Science Foundation of China (31770940 to YW and 81970536 to YW), the Natural Science Foundation of Beijing Municipality (7192097 to YW), and China Scholarship Council Grant (201706010305 to WL). This project has also been supported in part by Federal funds from the National Cancer Institute (NCI), National Institutes of Health (NIH), under Contract No. HSN261200800001E, and by the Intramural Research Programs of the NCI, CCR, CIP.

## Author contributions

W.L., Y.W., and J.M.W. designed the study. W.L. and X.P. performed most of the experiments and analysis. Q.L., P.L., Q.S., S.S., S.H., T.Y., W.G., and K.C. performed some experiments. P.W. performed GSEA analysis. W.Y., V.T., C.O., and G.T. provided experiments, key reagents, and analysis on microbiome sequencing. J.H., S.L., X.Y. and X. B. performed human specimen analysis. W.K. and J.X. designed and performed the generation of *Fam3D*$^{-/-}$ mice. W.L., J.M.W., and Y.W. prepared the manuscript with input from all authors.

## Competing interests

The authors declare no competing interests.
