## [Peer Review File · Nature Communications]

Reviewers' comments:

Reviewer #1 (Remarks to the Author); expert in colon cancer, inflammation and microbiome:

The manuscript „A novel cytokine like molecule, FAM3D, is essential for colon homeostasis and host defense against inflammation associated carcinogenesis“ by Liang et al. describes the impact of FAM3D a novel chemotactic agonist for FPR1 and 2 on gut homeostasis, colitis and inflammation-associated colorectal carcinogenesis.

Liang et al. show that FAM3D is mainly expressed in the gastrointestinal tract. Further they show that in the gut FAM3D is involved in intestinal homeostasis due to maintenance of mucosal integrity, regulation of epithelial cell and goblet cell proliferation, microbiome stability regulation and expression of antimicrobial peptides. Liang et al. propose that FAM3D play a critical role during pathogenesis of inflammatory bowel disease and colorectal cancer. Whereas the immune responses during intestinal homeostasis, IBD and CRC are studied in depth, the relationship between intestinal immune responses and its impact on bacterial dysbiosis are less clear. Therefore, it is critical to identify mechanisms that maintain gut homeostasis.

The study of Liang et al. is of major interest in the field of gastroenterology as there is a major interest in understanding intestinal homeostasis and regulation of the intestinal microbiota. The manuscript describes an important novel cytokine like molecule FAM3D that is involved in gut homeostasis, IBD and CRC pathogenesis. The manuscript at the moment is written a bit confusing and critical experiments are missing. Up to date the manuscript is not ready to publish in Nature Communications; however the subject of the article in general is suitable for publication in Nature Communications.

Major comments

- I suggest to rephrase all subtitles of the Results section. They are describing just the results of a specific experiment. I suggest rephrasing them to the conclusion which one can draw from the result of this experiment.
- I suggest to restructure the Result part. I think it would be logical to describe the function of FAM3D in intestinal homeostasis first. Then move on with IBD and CRC.
- A translational study of FAM3D expression in human IBD and CRC patients and its consequence for clinical parameters like prognosis, survival, and disease severity is missing and should be added.
- In general the quality of the immunohistochemistry figures is low. Mostly higher magnifications should be added either in the Figures or in the Supplementary Figures.
- Line 256 the authors draw the conclusion that Fam3D is constitutively expressed in IECs with villus and crypt expression pattern. The quality of the immunohistochemistry itself but also the picture quality and the magnification is not appropriate to draw this conclusion. Here, additional studies should be added like performing FAM3D Western Blot on isolated, purified IECs of the colon to be sure that FAM3D is expressed by IECs. Further a WB should be performed with intestinal organoids.
- Line 272 and Supplemental Figure 1C the TUNEL staining is looking really unspecific and not like a common TUNEL staining. I am not convinced that it has worked at all.
- Line 274 and Figure 1K from these immunofluorescence picture one cannot draw the conclusion of enhanced infiltration of macrophage, neutrophils and T cell infiltration because one can see absolute nothing. Maybe a higher magnification is suitable
- Line 279 – 293 and Figure 2E, there is missing a link to this Figure in the text. However, you can't see hyperplastic foci and neoplastic foci with this magnifications.
- Line 289 and Figure 2J, one cannot conclude from this HE staining any invasiveness of the adenocarcinomas. In this picture I cannot detect any invasive adenocarcinomas. I can detect tumors but in my eyes they are not invasive at the chosen pictures.
- Line 293 and Figure 2J panel d again I cannot see the increased β -catenin staining in this

pictures

- Regarding the tumor part, there is less information given about impact of Fam3D on tumorigenesis. I would suggest to study tumorigenesis in an spontaneous model of CRC and describe the mechanism in more depth. Therefore, I suggest omitting the whole tumor part and studying it in more depth for an additional manuscript. Here, I would concentrate more on homeostasis and IBD.
- Line 301, Suppl. Fig 2D in my eyes FAM3D is more trapped in the mucus than in IECs
- line 358 – 365 here a more depth analysis of the stem cell compartment of FAM3D KO mice should be performed especially a ISH staining of Lgr5 should be performed, but also the expression of further stem cell markers would be of interest.
- Line 366 Figure 5C please insert higher magnifications one can absolutely see no B cells, T cells or macrophages at these IF pictures
- The discussion in general is too superficial and the results and conclusions should be discussed in more depth.

Minor comments

- Line 308 here a link to Figure 3D is missing
- Line 366 Figure 5C shows a low level.... Should it not be high levels... also at line 370

Reviewer #2 (Remarks to the Author); expert in colitis mouse models and microbiome:

The authors have shown that FAM3D, a novel cytokine like molecule, is playing an essential role in colon homeostasis and host defense against inflammation associated carcinogenesis. They have demonstrated that the deficiency of FAM3D induced impaired integrity of the mucosa, increased epithelial hyper-proliferation, reduced anti microbial peptide production and increased susceptibility to colitis associated to high incidence of cancer. Moreover, FAM3D is suggested to impact the microbiota composition as the pretreatment of FAM3D mice with antibiotics reduced the severity of colitis. This observation indicated a critical role of the microbiota in mediating colitis in FAM3D mice. They have also transferred via co housing FAM3D microbiota to wt mice and thus confirmed the impact of the microbiota in the FAM3D mice phenotype.

Specific comments.

1. Concerning the link between FAM3D mice phenotype and their microbiota, the co housing experiments are nicely designed but it would have been interesting to confirm that FAM3D phenotype is associated to a dysbiotic treatment using germfree mice.
2. The use of Adenovirus to rectally deliver FAM3D in defective mice is also convincing but the authors do not use their interesting results to suggest potential translational impact as delivery of FAM3D using recombinant bacteria.
3. The discussion is well written but it is a bit disappointing that the authors do not propose any further strategies in Humans in order to validate their pre-clinical results. To conclude by a sentence confirming that FAM3D is a new class of guardians of the gut does not open the way to potential novel therapeutic strategies for IBD patients for example.
4. The other point that should be improved is the microbiota analysis which is quite descriptive in the Results part and even not commented in the discussion. They just identified the dysbiosis with an increase of Lactobacillus and decrease of Enterococcus faecalis among other bacteria. The authors do not exploit these observations and do not suggest any strategy based on the complementation of the dysbiotic microbiota with E. faecalis for example.

Philippe Langella, PhD, Head of ProbiHôte Lab in Micalis Institute (INRA, Jouy en Josas, France).

Point-by-point Responses

Reviewers' comments:

Reviewer #1

Major comments

1. I suggest to rephrase all subtitles of the Results section. They are describing just the results of a specific experiment. I suggest rephrasing them to the conclusion which one can draw from the result of this experiment.

Response: Thanks for the suggestion. We have made sure that all subtitles are clear to give a conclusive summary of results drawn from experiments. These subtitles are now listed below that initiate each paragraph in the Results section:

- High level expression of FAM3D (mouse Fam3D) in mouse gastrointestinal tract
- Increased spontaneous colitis and global inflammation in *Fam3D*^{-/-} mice
- Reduced homeostatic molecules in the colon epithelium of *Fam3D*^{-/-} mice
- Defective mucosal integrity in *Fam3D*^{-/-} mouse colon
- Increased severity of DSS-induced acute colitis in *Fam3D*^{-/-} mice
- Increased DSS-induced chronic colitis and AOM/DSS-induced carcinogenesis in *Fam3D*^{-/-} mice
- Rescue of established colitis in mice by Adv-FAM3D infusion
- Microbiota-driven spontaneous and DSS-induced colitis in *Fam3D*^{-/-} mice
- Altered composition of microbiota in the colon of *Fam3D*^{-/-} mice
- Transfer of *Fam3D*^{-/-} mouse phenotype to WT mice
- Significant reduction in FAM3D expression by human CRC

2. I suggest to restructure the Result part. I think it would be logical to describe the function of FAM3D in intestinal homeostasis first. Then move on with IBD and CRC.

Response: Thanks for the suggestion. We have restructured the Results part as suggested i.e. colon homeostasis first followed by IBD and CRC (see response to the above issue).

3. A translational study of FAM3D expression in human IBD and CRC patients and its consequence for clinical parameters like prognosis, survival, and disease severity is missing and should be added.

Response: We have examined the potential clinical relevance of our current findings. The clinical relevance of FAM3D expression to human colon cancer progression was demonstrated by analysis of TCGA (The Cancer Genome Atlas) and CPTAC (Clinical Proteomic Tumor Analysis Consortium) data in which FAM3D mRNA and protein are more highly expressed in normal human colon tissues than in CRC tissues (**Supplemental Figure 11a and b**) (**Ref #28, 29** in the **revised manuscript**). The analysis also revealed that FAM3D protein was expressed at the lowest level in stage IV cancers (**Supplemental Figure 11c and d**). These results are in accordance with our observations from animal studies, where deficiency in *Fam3D* resulted in the disruption of colon homeostasis accompanied by increased carcinogenesis. Therefore, FAM3D (*Fam3D*) clearly represents a novel anti-cancer molecule in both human and mice. A new paragraph has been included in the revised manuscript on **Page 26, Lines 582-592** with an additional figure as **Supplemental Figure 11** in the revised manuscript. The expression of FAM3D in human IBD was less conclusive in our studies of a limited number of specimens, with planned expansion in our future studies.

4. In general the quality of the immunohistochemistry figures is low. Mostly higher magnifications should be added either in the Figures or in the Supplementary Figures.

Response: Thanks for the suggestion. We have modified most of the immunohistochemistry figures with higher resolution. We also increased magnifications of some pictures, i.e. **Figure 1c, Figure 1h, Figure 3h, Figure 4e, Figure 4j, Supplemental Figure 1b, Supplemental Figure 7a** and **Supplemental Figure 8**.

5. Line 256 the authors draw the conclusion that *Fam3D* is constitutively expressed in IECs with villus and crypt expression pattern. The quality of the immunohistochemistry itself but also the picture quality and the magnification is not appropriate to draw this conclusion. Here, additional studies should be added like performing FAM3D Western Blot on isolated, purified IECs of the colon to be sure that FAM3D is expressed by IECs. Further a WB should be performed with intestinal organoids.

Response: Thanks for the advice. We have made corrections in the magnification of immunohistochemistry pictures. As shown in **Figure 1c** and **Supplemental Figure 1b** in the revised manuscript, higher magnification figures revealed that Fam3D was constitutively expressed in colon epithelial cells with a prominent localization in villus and crypts. We isolated IECs from normal mouse colon and found abundant expression of Fam3D as measured by real-time PCR (**Figure 1d**), which is consistent with high levels of Fam3D protein in colonic IECs as confirmed by Western blot (**Figure 1e**). Also, dual immunofluorescent staining detected the production of Fam3D by colonic epithelial cells (EpCAM-positive cells) (**revised Supplemental Figure 1d**), and **Page 17, Lines 361-365**.

In intestinal organoids, we detected Fam3D in crypt structures that recapitulated native intestinal epithelium, with higher expression level in the organoids incubated with DAPT, a Notch inhibitor, to induce the differentiation of cultured Lgr5⁺ intestinal stem cells as measured by Western blot and confocal microscopy (**Supplemental Figure 1e and f**). We therefore confirmed that Fam3D is constitutively expressed by colonic epithelial cells of the WT mice (**Page 17, Lines 366-370**).

6. Line 272 and Supplemental Figure 1C the TUNEL staining is looking really unspecific and not like a common TUNEL staining. I am not convinced that it has worked at all.

Response: We have re-stained samples carefully and a new figure was presented in the revised manuscript with more convincing TUNEL staining (**revised Supplemental Figure 7a, right panel**).

7. Line 274 and Figure 1K from these immunofluorescence picture one cannot draw the conclusion of enhanced infiltration of macrophage, neutrophils and T cell infiltration because one can see absolute nothing. Maybe a higher magnification is suitable.

Response: Thanks for the comment. We have included a new figure in the revised manuscript (**Figure 3h**). In addition, we isolated lamina propria of WT and *Fam3D*^{-/-} mouse colonic tissues with chemically induced colitis and stained the tissues with relevant antibodies to show inflammatory cells. Compared with WT mouse colon, there was a marked increase in myeloid cells (neutrophils and monocytes/macrophages), T cells and B cells in the lamina propria of

Fam3D^{-/-} mouse colon (**Figure 3h**). FACS analysis results are shown in the revised Supplementary Materials (**Supplemental Figure 7b and c**, and **Page 21, Lines 468-471**).

8. Line 279 – 293 and Figure 2E, there is missing a link to this Figure in the text. However, you can't see hyperplastic foci and neoplastic foci with this magnifications.

Response: Thanks for pointing out the error. We have cited images with enhanced magnification as **revised Figure 4e**, which showed marked leukocyte infiltration and extensive hyperplastic and neoplastic foci, especially near the mucosal surface in the colon of *Fam3D*^{-/-} mice (**Figure 4e**). For further characterization of pathologic changes in DSS-induced chronic colitis in *Fam3D*^{-/-} mice, we stained colon with Ki67 and confirmed the presence of hyperproliferative cells in lesions as compared with adjacent non-neoplastic tissues (**Supplemental Figure 8a**). These results indicate impaired repair of the colonic epithelium after injury in *Fam3D*^{-/-} mice with increased tumorigenesis. The results are described on **Page 22, Lines 480-485** in the revised manuscript.

9. Line 289 and Figure 2J, one cannot conclude from this HE staining any invasiveness of the adenocarcinomas. In this picture I cannot detect any invasive adenocarcinomas. I can detect tumors but in my eyes they are not invasive at the chosen pictures.

Response: Thanks for the comment. We now included new images with enhanced magnification in the **revised Figure 4j, Panel (i)** (**Page 22, Lines 489-493**). The colorectal cancer in *Fam3D*^{-/-} mice progressed more rapidly than the tumors in WT mice as shown by more invasive tumor nodules, with deeper invasion into the submucosal region in morphology of adenocarcinomas with atypical glandular architectures in the tumors (**Figure 4j, Panel (i)** and **Supplemental Figure 8b**).

10. Line 293 and Figure 2J panel d again I cannot see the increased β -catenin staining in these pictures.

Response: Thanks for the reminder. We have included an improved figure of immunofluorescent staining for β -catenin in the revised manuscript (**Figure 4j, Panel (ii)** and **Supplemental Figure 8c**). **Figure 4j, Panel b** showed an increased level of β -catenin in proliferative crypts and invasive nodules in the colon of *Fam3D*^{-/-} mice. **Supplemental Figure**

8c showed submucosal nodules located in muscularis mucosa displaying few positive nuclei, with β -catenin localized on the membrane of epithelial cells without any nuclear clustering. These results indicate that *Fam3D* deficiency promotes inflammation-associated carcinogenesis in mouse colon, involving increased Wnt signaling in transformed epithelial nodules as evidenced by elevated β -catenin (Page 22, Lines 493-499).

11. Regarding the tumor part, there is less information given about impact of Fam3D on tumorigenesis. I would suggest to study tumorigenesis in a spontaneous model of CRC and describe the mechanism in more depth. Therefore, I suggest omitting the whole tumor part and studying it in more depth for an additional manuscript. Here, I would concentrate more on homeostasis and IBD.

Response: The Reviewer's suggestion is appreciated. However, we believe that tumorigenesis is a critical consequence of increased inflammatory responses in the colon of *Fam3D*^{-/-} mice, which is highly relevant to the theme of our current study. Therefore, the carcinogenesis part is retained in the revised manuscript. The inclusion of the tumor results was also endorsed by Editor's recommendation.

12. Line 301, Suppl. Fig 2D in my eyes FAM3D is more trapped in the mucus than in IECs

Response: Thanks for the comment. Based on the Reviewer's comment, we have modified our statement on Page 23, Lines 506-507 as "We then infused Adv-FAM3D into the rectum of mice and detected the expression of FAM3D in colon tissues".

13. line 358 – 365 here a more depth analysis of the stem cell compartment of FAM3D KO mice should be performed especially a ISH staining of Lgr5 should be performed, but also the expression of further stem cell markers would be of interest.

Response: Based on the Reviewer's suggestion, we have analyzed the stem cell compartment of *Fam3D*^{-/-} colon. There is no difference in the numbers of Lgr5⁺ cells in the colon of WT and *Fam3D*^{-/-} mice as measured by immunofluorescent staining (Supplemental Figure 2a), or stem cell related genes, such as *Axin2*, *Olfm4*, *Smoc2*, *Msi1*, *Ascl2* etc. as measured by RNA-seq (Supplemental Figure 2b and c). Therefore, *Fam3D* deficiency did not affect the generation of

stem cells. We further analyzed stem cell compartment by using colon organoid culture. The formation and size of organoids from isolated colon crypts did not show difference between WT and *Fam3D*^{-/-} mice (**Supplemental Figure 2d and e**). These results confirm that Fam3D is not directly involved in the renewal of colonic mucosal layer. The results are described on **Page 18, Lines 383-391** in the revised manuscript.

14. Line 366 Figure 5C please insert higher magnifications one can absolutely see no B cells, T cells or macrophages at these IF pictures

Response: Thanks for the suggestion. We have included images with higher magnification in **revised Figure 1h** with descriptions on **Page 18, Lines 392-395**. Analysis of colonic lamina propria leukocytes in 1-year old WT and *Fam3D*^{-/-} mice by using FACS showed increased neutrophils and monocytes/macrophages in the colon of *Fam3D*^{-/-} mice, with no significant difference in T and B cells. The results are shown in the revised Supplementary Materials (**Supplemental Figure 3a-b** with descriptions on **Page 19, Lines 400-403**).

15. The discussion in general is too superficial and the results and conclusions should be discussed in more depth.

Response: We have added three new paragraphs in the **Discussion** section to accommodate the Reviewer's suggestion (**Page 27, Lines 606-617; Page 29-31, Lines 658-698**).

Minor comments

16. Line 308 here a link to Figure 3D is missing

Response: We are sorry for the missing link. We now cited **Figure 5d** in the revised manuscript on **Page 23, Line 516**.

17. Line 366 Figure 5C shows a low level.... Should it not be high levels... also at line 37

Response: Thanks for pointing this out. We have corrected the statement in the revised manuscript on **Page 18, Lines 393 and 395**.

Reviewer #2 (Remarks to the Author):

General comments:

Response: We thank the reviewer for positive assessment of our manuscript.

Specific comments.

- 1. Concerning the link between FAM3D mice phenotype and their microbiota, the co-housing experiments are nicely designed but it would have been interesting to confirm that FAM3D phenotype is associated to a dysbiotic treatment using germfree mice.**

Response: Thanks for the advice. Several lines of evidence support the role of Fam3D in maintaining normal colon microbiome: (1) There was an abnormal composition of microbiome in the colon of *Fam3D*^{-/-} mice (**Figure 6f-j, Supplemental Figure 10a-c**); (2) The elongation of colon crypts and an increased susceptibility to chemically induced colitis shown by *Fam3D*^{-/-} mice were reduced by antibiotic treatment (**Figure 6a-d**); (3) WT mice cohoused with *Fam3D*^{-/-} mice adopted the phenotype of crypt elongation and increased sensitivity to chemically induced colitis (**Figure 7b-f**); (4) Separation of mice after cohousing confirmed the dysbiosis associated with *Fam3D* deficiency (**Figure 7a, Supplemental Figure 10d-e**); and furthermore; (5) Single-housing of mice derived from heterozygous breeders demonstrated significant divergence in microbiome between WT and *Fam3D*^{-/-} mice (**Supplemental Figure 10f-h**).

We agree with the Reviewer that germ-free approach may provide a clearer answer. We therefore set out to establish germ-free (GF) colonies of both WT and *Fam3D*^{-/-} mice. However, it is unfortunate that the approach is very time consuming and extremely expensive. Our effort to establish germ free colonies on the campus (now at a re-derivation stage) also encountered further difficulties due to COVID-19 pandemic that by mandates all research activities have been under strict restriction. In the meantime, we believe germ free approach per se may constitute an independent project that involves numerous examinations of naïve as well as stimulated states of the mice, plus microbiome transfer. However, we believe the evidence we obtained thus far is adequate to justify our conclusions about the role of microbiome in the phenotype of *Fam3D*^{-/-} mice. Nevertheless, if the GF mouse studies are deemed absolutely required, we would like to request an extension of the deadline so that we may at least obtain certain preliminary results.

- 2. The use of Adenovirus to rectally deliver FAM3D in defective mice is also convincing but the authors do not use their interesting results to suggest potential translational impact as delivery of FAM3D using recombinant bacteria.**

Response: Thanks for the advice. Fam3D deficiency impaired mucosal barrier function of the colon with reduced expression of acidic mucins, a thinner mucus layer and reduced antimicrobial peptides (AMPs), which were associated with gut dysbiosis. Fam3D thus constitutes a novel therapeutic agent as demonstrated by the capacity of adenoviral delivered Fam3D to substantially rescue the abnormalities shown by *Fam3D*^{-/-} mice. In addition, our studies of human specimens indicate that Fam3D was highly expressed in normal human colon tissues with substantial decrease in CRC tissues at both mRNA and protein levels (**Supplemental Figure 11a-d**). Thus, FAM3D also exhibits its capacity to protect human colon from carcinogenesis. It is therefore particularly promising and important that FAM3D be administered in human for management of colitis and associated cancer by using safer and reliable recombinant bacterial delivery systems. These statements have been included in the **Results (Last paragraph, Page 26, Lines 582-592)** and **Discussion (Page 30, Lines 683-698)** sections of the **revised manuscript**.

3. The discussion is well written but it is a bit disappointing that the authors do not propose any further strategies in Humans in order to validate their pre-clinical results. To conclude by a sentence confirming that FAM3D is a new class of guardians of the gut does not open the way to potential novel therapeutic strategies for IBD patients for example.

Response: We agree with the Reviewer. We now state in the revised manuscript that such an approach may be useful for treatment of IBD and CRC in human. Fam3D is essential for colon homeostasis as manifested by observations in which its deficiency results in abnormal development of colon goblet cell compartment, and dysbiosis-associated inflammation and tumorigenesis. In human, FAM3D was decreased in CRC (**Supplemental Figure 11a-d**) and adenoviral delivered of FAM3D was able to rescue the colitis seen in the gut of both WT and *Fam3D*^{-/-} mice, suggesting the potential of using FAM3D to alleviate the severity of human colitis. Therefore, FAM3D (Fam3D) clearly represents a novel anti-cancer molecule in both human and mice.

A new paragraph has been included in the revised manuscript on **Page 30, Lines 683-698** in the revised manuscript.

4. The other point that should be improved is the microbiota analysis which is quite descriptive in the Results part and even not commented in the discussion. They just identified the dysbiosis with an increase of Lactobacillus and decrease of Enterococcus faecalis among other bacteria. The authors do not exploit these observations and do not suggest any strategy based on the complementation of the dysbiotic microbiota with E. faecalis for example.

Response: Thanks for the suggestion. We agree with the Reviewer that the abnormal microbiota should be exploited further for its pathophysiological implications. We now have added the following statement in the **Discussion** section of the **revised manuscript (Page 29, Lines 658-682)**.

Our study revealed that in *Fam3D*^{-/-} mice, several bacterial strains were expanded including potential pathogenic strains *Muribaculaceae* and *Deferribacteraceae*, with reduction in the abundance of beneficial commensal species including *Ruminococaceae*, which is an important short fat chain producer (**Ref#50**), and Segmented filamentous bacteria, as an antibacterial peptide inducer (**Ref#51**). Our observations are consistent with animal models that exhibit colon mucosal barrier dysfunction such as *Muc2* and *Nod2* deficiency (**Ref#52, 53**). In addition, a large proportion of bacterial species expanded in *Fam3D*^{-/-} mice are related to mucus degradation thus being colitogenic including *Peptostreptococcus* and *Clostridium perfringens*. *Peptostreptococcus*, also more abundant in *Fam3D*^{-/-} mice, is significantly enriched in patients with UC and CRC (**Ref#54**). Further, *Clostridium perfringens* exerts proteolytic and mucinase activity, implicated in the pathogenesis of IBD (**Ref#55**), related to a thinner mucus layer necessary for protection of colonic mucosa. Thus, dysbiosis demonstrated in the colon of *Fam3D*^{-/-} mice is correlated with their increased severity of inflammation and tumorigenesis.

However, *Lactobacillus/Lactococcus* and *Enterococcus faecalis* (*E. faecalis*) showed less definable changes in the colon of *Fam3D*^{-/-} mice. As a critical ingredient of probiotics, *Lactobacillus/Lactococcus* has shown the capacity to maintain homeostasis of the gut and as a treatment for IBD (**Ref#56, 57**). *E. faecalis*, as a proinflammatory bacteria, is found to correlate strongly with the progression of IBD (**Ref#58**). Similar intestinal bacterial pattern has been described in *Nod2*^{-/-} mice with elimination of anaerobic bacteria (**Ref#53**), suggesting

abnormally increased *Lactobacillus* might be a result of imbalanced composition of commensal microbiota as a sign of dysbiosis.

These results highlight the complexity of microbiota that undergoes dynamic changes with the microenvironment, therefore further studies with complementation of dysbiotic microbiota should be warranted. Regardless, we are still at the beginning of fully deciphering the contribution of Fam3D to microbiota balance and the opportunity to take the advantage of available information as novel therapeutic approaches which belong to our future study plans.

REVIEWERS' COMMENTS:

Reviewer #3 (Remarks to the Author):

The revised manuscript by Liang and colleagues on the role of FAM3D in the intestine seems to have addressed most of the other reviewers' initial concerns. I have not seen the previous manuscript, however, while the data clearly suggest a role of FAM3D in intestinal homeostasis and DSS induced colitis, the underlying mechanism explaining the observed phenotype is lacking. The authors describe a phenotype that affects both mucosal integrity as well as the composition of the intestinal microbiome, yet it remains obscure how the DSS induced colitis is affected by these and how particularly the viral administration of FAM3D may improve colitis. Moreover, the therapeutic effect is very limited. It remains also unclear whether a cell autonomous effect is responsible or the changes in the microbiome.

Given the wide variety of changes in the intestinal mucosa upon FAM3d loss, the explanations for the observed phenotypes could be manifold, however, not a single one is followed up or examined and as such the manuscript provides only correlative and descriptive data.

Reviewer #4 (Remarks to the Author):

The authors have addressed most of the comments. There are still issues with the new added text in the discussion that requires attention:

1) This following sentence is not making sense, especially the first section.

"As commensal bacteria rely on intestinal homeostasis as a habitant environment, it was plausible that significant differences in microbial communities seen in Fam3D^{-/-} are associated with pathophysiological changes in the colon."

2) Please correct "short fat chain producer" to "short chain fatty acid".

"These changes include the expansion of potential pathogens including Muribaculaceae and Deferribacteraceae, and shrinking of beneficial commensal species i.e Ruminococaceae, as an important short fat chain producer⁵⁰, and Segmented filamentous bacteria, as an antibacterial peptide inducer⁵¹."

3) This sentence needs to be modified. Bacterial species are not "up-regulated" in an ecosystem. The authors probably refer to "increased relative abundance". Please be careful with the microbiome language.

"Additionally, a large proportion of bacterial species that were up-regulated in Fam3D^{-/-} gut are also related to mucus degradation thus considered as colitogenic including Peptostreptococcus and Clostridium perfringens. Peptostreptococcus, more abundant in Fam3D^{-/-} mice, is also significantly enriched in patients with UC and 30 CRC⁵⁴."

4) This sentence uses a reference for a disease called pouchitis, not IBD. The bacterium is a potential predictor of pouchitis.

"Also, Clostridium perfringens exerts proteolytic and mucinase activity, both of which may play a role in the pathogenesis of IBD⁵⁵, and may be attributed in part to a thinner mucus layer

observed in the colon of Fam3D^{-/-} mice.”

5) The paragraph mentioned below is misleading. First, they are not effective probiotics against human IBD. Second, genus-level microorganisms are quite different than species/subspecies in terms of function. One can't equate genus to a specific function regarding probiotics. Second, the comment about *E. faecalis* is incorrect since the reference indicates a study performed in mice. Please re-write with more rigor and accuracy.

“In contrast, *Lactobacillus/Lactococcus* and *Enterococcus faecalis* (*E. faecalis*) showed unpredictable changes in the colon of Fam3D^{-/-} mice. As a critical ingredient of probiotics, *Lactobacillus/Lactococcus* have been described to maintain homeostasis of gut and as a treatment for IBD^{56,57}. *E. faecalis*, as a proinflammatory bacterium, is found to correlate strongly with IBD status⁵⁸.”

6) Although this reviewer appreciates the effort made by the authors to connect their study to clinical application, one needs to recognize the limitation of the current study. The acute DSS model is not representative of IBD, and therefore the role of FAM3D should first be investigated in a T cell model of chronic intestinal inflammation before moving this potential target into clinical trials. Please tone down this paragraph to highlight model limitation before going to clinical arena.

“These findings may have clinical relevance since the human counterpart protein FAM3D was more highly expressed in normal colon tissues in contrast to CRC tissues with the lowest levels of both FAM3D mRNA and protein expression in Grade IV CRC, with which the patients suffer from the poorest survival. Thus, the impact of Fam3D on colon homeostasis and diseases is multifaceted and its capacity to protect normal colon should be exploited as a future therapeutic modality, as already revealed in our study that anal administration of adenovirus-encoded FAM3D significantly improved the severity of colitis in mice. It is therefore particularly promising and important that FAM3D be administered in humans for management of colitis and associated cancer by using safer and reliable recombinant bacterial delivery systems.”

Point-by-point reply to the comments by Reviewers #3 and #4

Reviewer 3

The Reviewer expressed concerns about the “correlative and descriptive” nature data of the data presented in our manuscript.

Response: We appreciate the comments. We further discussed the mechanistic basis for the role of FAM3D in colon homeostasis obtained so far in our study. We have demonstrated that in the present manuscript that in the homeostatic state, absence of Fam3D is associated with impaired mucosal barrier function with abnormal development of goblet cells (GCs), decreased production of antimicrobial peptides (AMPs) by colon epithelial cells, reduced thickness of the mucus layer, as well as a mild spontaneous colitis associated with epithelium hyperplasia. This spontaneous phenotype constitutes the basis for dysbiosis, aggravated chemically induced colitis and carcinogenesis. More in depth experiments revealed that *Fam3D* deficiency is associated with the lack of acidic modification of mucins and mucus secretion therefore their inability to form a continuous inner mucus layer on the colon mucosa contributing to the skewed balance of microbiome culminating in the increased susceptibility of the mice to colitis and carcinogenesis. Further study is ongoing to clarify the nature of cellular receptors or sensors for Fam3D to exert its critical function in the colon. We now have included the above statement in the **Discussion** section of the revised manuscript (**Page 18, Lines 384-391**).

Reviewer #4:

1) This following sentence is not making sense, especially the first section.

“As commensal bacteria rely on intestinal homeostasis as a habitant environment, it was plausible that significant differences in microbial communities seen in Fam3D^{-/-} are associated with pathophysiological changes in the colon.”

Response: Thanks for the comments. We have modified the sentence as “As intestinal homeostasis is critical for the balance of normal microbiome in the colon, it is plausible that the dysbiosis seen in *Fam3D*^{-/-} mice is a consequence of pathological changes occurred in the colon” on **Page 18, Line 392-394**.

2) Please correct “short fat chain producer” to “short chain fatty acid”.

“These changes include the expansion of potential pathogens including Muribaculaceae and Deferribacteraceae, and shrinking of beneficial commensal species i.e Ruminococaceae, as an important short fat chain producer⁵⁰, and Segmented filamentous bacteria, as an antibacterial peptide inducer⁵¹.”

Response: Thanks for pointing out the error. “Short fat chain” has been corrected to “short chain fatty acid” on **Page 18, Line 396**.

3) This sentence needs to be modified. Bacterial species are not “up-regulated” in an ecosystem. The authors probably refer to “increased relative abundance”.

Response: Thanks for the modification. We have modified the words “up-regulated” to “increased relative abundance” as suggested by the Reviewer (**Page 18, Line 399**).

4) This sentence uses a reference for a disease called pouchitis, not IBD. The bacterium is a potential predictor of pouchitis.

Response: Thanks for the correction. We have corrected the statement to “Also, *Clostridium perfringens* exerts proteolytic and mucinase activity, both of which may play a role in the pathogenesis of pouchitis, a colon inflammatory disease, and may be attributed in part to a thinner mucus layer observed in the colon of *Fam3D*^{-/-} mice” on **Page 19, Line 404**.

5) The paragraph mentions below is misleading. First, they are no effective probiotics against human IBD. Second, genus level microorganisms are quite different than species/subspecies in term of function. One can’t equate genus to a specific function regarding probiotics. Second, the comment about *E. faecalis* is incorrect since the reference indicates a study performed in mice. Please re-write with more rigor and accuracy.

Response: Thanks for the advice. We have re-written the statement as follows: “In contrast, *Lactobacillus/Lactococcus* and *Enterococcus faecalis* (*E. faecalis*) showed unpredictable changes in the colon of *Fam3D*^{-/-} mice. Some *Lactobacilli/Lactococcus* strains are potential probiotics as they can maintain gut homeostasis. *E. faecalis*, as a proinflammatory bacterium, is

found to correlate strongly with murine colitis, with yet to be confirmed relevance to human IBD.” on Page 19, Lines 407-410.

6) Although this reviewer appreciates the effort made by the authors to connect their study to clinical application, one needs to recognize the limitation of the current study. The acute DSS model is not representative of IBD, and therefore the role of FAM3D should first be investigated in T cell model of chronic intestinal inflammation before moving this potential target into clinical trials. Please tone down this paragraph to highlight model limitation before going to clinical arena.

Response: Thanks for the suggestion. We have toned down the statement as follows: “It is therefore promising that FAM3D may be administered in human for management of colitis and associated cancer by using safer and reliable recombinant bacterial delivery systems. However, DSS-induced colitis in mice has limited representation to human IBD, therefore more relevant models such as T-cell-transferred chronic colitis should be considered in future studies to clarify the significance of FAM3D in the pathogenesis of human IBD and associated cancer as well as a as therapeutic agent.” (Page 20, Lines 432-437)

In summary, we are very grateful to the Reviewers and the Editors for the direction in elevating the levels of our scientific research. We hope our responses and revisions are considered satisfactory to warrant the publication of the manuscript.